biomechanics, physiology

biomechanics, muscle-tendon unit,
force–length–velocity relationships, gear ratio,
running economy

**Author for correspondence:**
Sebastian Bohm
e-mail: sebastian.bohm@hu-berlin.de

# The force–length–velocity potential of the human soleus muscle is related to the energetic cost of running

Sebastian Bohm[1,2], Falk Mersmann[1,2], Alessandro Santuz[1,2] and Adamantios Arampatzis[1,2]

[1]Department of Training and Movement Sciences, and [2]Berlin School of Movement Sciences, Humboldt-Universität zu Berlin, Berlin, Germany

  SB, 0000-0002-5720-3672; FM, 0000-0001-7180-7109; AS, 0000-0002-6577-5101;
AA, 0000-0002-4985-0335

According to the force–length–velocity relationships, the muscle force potential is determined by the operating length and velocity, which affects the energetic cost of contraction. During running, the human soleus muscle produces mechanical work through active shortening and provides the majority of propulsion. The trade-off between work production and alterations of the force–length and force–velocity potentials (i.e. fraction of maximum force according to the force–length–velocity curves) might mediate the energetic cost of running. By mapping the operating length and velocity of the soleus fascicles onto the experimentally assessed force–length and force–velocity curves, we investigated the association between the energetic cost and the force–length–velocity potentials during running. The fascicles operated close to optimal length ($0.90 \pm 0.10$ $L_0$) with moderate velocity ($0.118 \pm 0.039$ $V_{max}$ [maximum shortening velocity]) and, thus, with a force–length potential of $0.92 \pm 0.07$ and a force–velocity potential of $0.63 \pm 0.09$. The overall force–length–velocity potential was inversely related ($r = -0.52$, $p = 0.02$) to the energetic cost, mainly determined by a reduced shortening velocity. Lower shortening velocity was largely explained ($p < 0.001$, $R^2 = 0.928$) by greater tendon gearing, shorter Achilles tendon lever arm, greater muscle belly gearing and smaller ankle angle velocity. Here, we provide the first experimental evidence that lower shortening velocities of the soleus muscle improve running economy.

## 1. Background

Humans are capable runners compared with most other mammals and it has been suggested that the endurance performance has been a crucial aspect for human evolution [1]. Running economy is an important physiological factor for endurance performance [2] and is defined as the mass-specific rate of oxygen uptake or metabolic energy consumption at a given speed [3,4]. The main determinant of the metabolic energy consumption during running is the muscular force needed to support and accelerate the body mass [5]. The level of muscle activation necessary to generate the required force is dictated by the force–length and force–velocity potential of the muscle. The force–length and force–velocity potential express the operating length and velocity of the muscle fibres with respect to the force–length [6] and force–velocity relationships [7] (i.e. fraction of maximum force according to the force–length–velocity curves) [8,9]. When fibres operate at lower shortening velocities and close to the optimal length, the required active muscle volume for a given force diminishes, together with the metabolic energy expenditure [4,10]. Besides the operating length and velocity as the main determinants, the history dependence of force generation (i.e. increased force after active muscle lengthening

[11] and decreased force after active shortening [12]) may additionally influence the force potential. Thus, it is reasonable to argue that the fibre dynamics of the large lower limb muscles during running are explanatory factors of the energetic cost and thereby endurance performance.

During human running, the soleus actively shortens [13] and is the most important muscle for propulsion [14,15]. However, during active shortening, increased length excursion and shortening velocity reduce the force–length–velocity potential of muscle fibres. Due to the steep slope of the hyperbolic force–velocity curve at low to moderate shortening velocities, the force–velocity potential might be particularly sensitive to changes in shortening velocity. Yet the association between energetic cost and operating fibre dynamics as the force–length and force–velocity potential during human running has not been experimentally investigated thus far.

From a mechanical point of view, the soleus fibre operating length and velocity are mainly mediated by the decoupling of the fibre length trajectories from those of the muscle–tendon unit (MTU), the Achilles tendon lever arm and the excursions of the ankle joint. The decoupling of the fibre length trajectories from the MTU is a result of tendon compliance and the variable pennation of muscle fibres within the series muscle belly and can be quantified by the so-called *MTU gearing* (i.e. ratio of MTU and fibre velocity) [16]. Tendons, due to their compliance, take over important portions of the length changes within the MTU, which substantially reduce the length change and velocity of the series muscle belly. The magnitude of the decoupling of the muscle belly from the MTU by the tendon is expressed by the ratio of MTU velocity and belly velocity and has been termed *tendon gearing* [16]. Furthermore, the rotation of the fibres (i.e. changes of pennation angle) during muscle shortening and concomitant changes in muscle shape decouple the fibre length change from the length change of the muscle belly, further decreasing the fibre shortening length and velocity [17]. The ratio of muscle belly velocity and fibre velocity defines the effect of the fibre rotation mechanism on the shortening velocity, i.e. *belly gearing* (or architectural gear ratio) [16,17]. Independent of the gearing within the soleus MTU, muscle force is transmitted through the Achilles tendon lever arm (i.e. the distance between the tendon's line of action and the centre of rotation of the ankle joint). It has been shown that shorter lever arms of the Achilles tendon are correlated with lower rates of energy consumption during running [18,19]. The lower energy consumption has been attributed to a greater energy storage and return by the Achilles tendon due to the higher muscle force required for a given joint moment at a smaller lever arm [18]. However, the increased energy storage and release from the tendon is associated with a higher muscle force, which in turn increases the metabolic cost, counteracting or even deteriorating the effects of increased energy storage and release [20]. Yet shorter lever arms can reduce the fibre length excursions and fibre shortening velocity of the soleus muscle at a given ankle joint excursion during the stance phase, which can increase the muscle force–length–velocity potential. Therefore, besides the debated benefits in terms of energy storage and release, a reduction of the fibre operating length changes and velocity by a shorter lever arm of the Achilles tendon could be an important mechanism for the improvement in running economy. Furthermore, the ankle joint excursion during the stance phase of running may also influence the operating length and velocity of the soleus fibres [21]. Although gearing within the MTU contributes to the decoupling of fibre length and MTU length trajectories, smaller ankle joint excursions can decrease fibre length changes and velocities. In fact, Cavagna & Kaneko [22] as well as Williams & Cavanagh [23] reported reduced ankle joint excursions in runners with higher running economy than others [22,23].

In the present study, we investigated the operating length and velocity of the soleus muscle fascicles (i.e. bundles of fibres) during running as a function of the experimentally determined force–length and assessed force–velocity relationships (i.e. force–length and force–velocity potential) and their association to the energetic cost of running. We further assessed tendon and belly gearing as well as Achilles tendon lever arm and ankle joint excursions during the stance phase of running as mediating factors for the fascicle operating length and velocity. We hypothesized the force–length–velocity potential to be associated with the energetic cost of running, mainly due to the sensitivity of the force–velocity potential to modulations of fascicle velocity. Finally, we expected that gearing, tendon lever arm and joint excursion would explain the majority of the fascicle velocity variability in the soleus muscle during running.

## 2. Methods

### (a) Experimental design

Nineteen healthy (age: $29 \pm 6$ years, height: $177 \pm 9$ cm, mass: $69 \pm 9$ kg, 7 female), ambitious runners who trained at least three times per week participated in the present study. The ethics committee of the Humboldt-Universität zu Berlin approved the study and the participants gave written informed consent in accordance with the Declaration of Helsinki.

After familiarization, the participants ran on a treadmill at $2.5$ m s$^{-1}$ for 4 min. By integrating ultrasonography, electromyography (EMG) and kinematic data, we measured muscle architectural parameters (fascicle length, pennation angle and thickness) and EMG activity, and assessed the MTU length of the soleus muscle as well as the ankle joint angle of the right leg. Energetic cost of running was determined by expired gas analysis during an additional 10 min running trial at the same speed. In the second part of the experiment, the individual force–fascicle length relationship of the soleus was experimentally determined by means of maximal isometric voluntary plantar flexion contractions (MVC) of the right leg at different ankle joint angles on a dynamometer in combination with ultrasound imaging of the soleus muscle fascicles. The force applied to the Achilles tendon was calculated from the ankle joint moment and the individual tendon lever arm. The derived optimal fascicle length for force production was further used to determine the force–velocity relationship of the soleus fascicles. The order of the two parts of the experiments (running and MVC) was randomized, yet the ultrasound probe and EMG electrodes remained attached between both ultrasound measurements. Based on the assessed force–length and force–velocity relationships, it was possible to calculate the force–length and force–velocity potential of the soleus muscle as a function of the fascicle operating length and velocity during the stance phase of running. The product of both potentials then gives the overall force–length–velocity potential.

### (b) Assessment of the soleus force–length and force–velocity relationship

The participants were placed in prone position on the bench of an isokinetic dynamometer (Biodex Medical, Syst. 3, Inc., Shirley, NY) with the knee in fixed flexed position (approx. 120°)

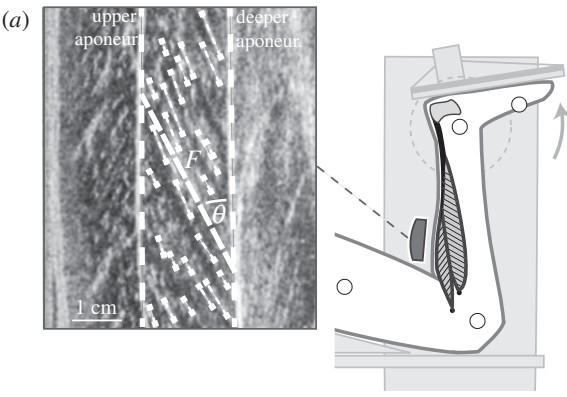

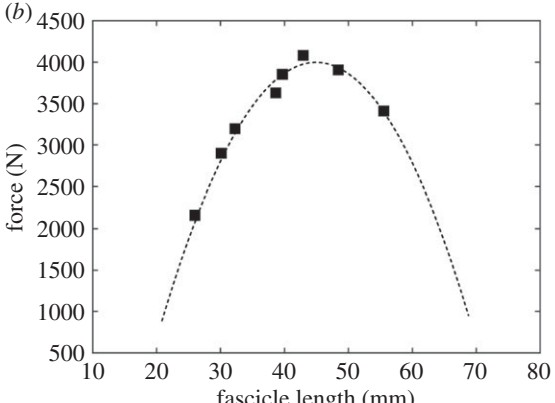

**Figure 1.** Experimental set-up for the determination of the soleus force–fascicle length relationship. (*a*) Maximum isometric plantar flexions (MVC) in eight different joint angles were performed on a dynamometer. During the MVCs, the soleus muscle fascicle length (*F*), pennation angle (*Θ*) and muscle thickness were measured based on ultrasound images. (*b*) Exemplary force–fascicle length relationship of the soleus muscle by the MVCs (squares) and the respective second-order polynomial fit (dashed line).

to restrict the contribution of the bi-articular m. gastrocnemius to the plantar flexion moment [24] (figure 1*a*). Following a standardized warm-up, MVCs were performed with the right leg in eight different joint angles, including a plateau of around 3 s. The angles ranged from 10° plantar flexion to the individual maximum dorsiflexion angle, set in random order in uniformly distributed intervals. The moments at the ankle joint were calculated taking into account the effects of gravitational and passive moments and any misalignment between ankle joint axis and dynamometer axis by means of an established inverse dynamics approach [25] as well as the contribution of the antagonistic muscles by means of electromyography (description in electronic supplementary material; figure 1*a*). The force applied to the Achilles tendon during the plantar flexion MVCs was calculated as quotient of the joint moment and the individual tendon lever arm (description in electronic supplementary material). Soleus fascicle behaviour during the MVCs was synchronously captured at 30 Hz by B-mode ultrasonography (Aloka Prosound Alpha 7, Hitachi, Tokyo, Japan) with a 6 cm linear array probe (UST-5713T, 13.3 MHz). The probe was mounted on the shank over the medial aspect of the soleus muscle belly by means of a custom made antiskid neoprene/plastic cast (figure 1*a*). The fascicle length was post-processed from the ultrasound images (figure 1*a*) using a self-developed semi-automatic tracking algorithm [26], described in more detail in the electronic supplementary material. Accordingly, an individual force–fascicle length relationship was calculated for each participant based on a second-order polynomial fit (figure 1*b*) and the maximum muscle force applied to the tendon ($F_{max}$) and optimal fascicle length for force generation ($L_0$) was derived, respectively. Furthermore, we assessed the force–velocity relationship of the soleus using the classical Hill equation

[7], and the muscle-specific maximum fascicle shortening velocity ($V_{max}$) and constants of $a_{rel}$ and $b_{rel}$. $V_{max}$ was derived from the study of Luden *et al.* [27], which showed $V_{max}$ values for type 1 fibres of 0.77 $L_0$ s$^{-1}$ and 2.91 $L_0$ s$^{-1}$ for type 2 fibres of the human soleus muscle measured *in vitro* at 15°C [27]. Considering the temperature coefficient [28], $V_{max}$ can be predicted as 4.4 $L_0$ s$^{-1}$ for type 1 fibres and 16.8 $L_0$ s$^{-1}$ for type 2 fibres under physiological temperature conditions (37°C). Using an average fibre type distribution (type 1 fibres: 81%, type 2: 19%) of the human soleus muscle reported in literature [27,29–31], $V_{max}$ can be calculated as 6.77 $L_0$ s$^{-1}$. $a_{rel}$ was calculated as 0.1 + 0.4FT, where FT is the fast twitch fibre type percentage (see above), which then equals to 0.175 [32,33]. The product of $a_{rel}$ and $V_{max}$ then gives $b_{rel}$ as 1.182 [34]. After rearrangement of the Hill equation and extension to the eccentric component, the operating velocity normalized to $V_{max}$ was used to calculate the individual force potential according to the force–velocity relationship.

## (c) Assessment of joint kinematics, muscle architecture and electromyographic activity during running

During running on a treadmill (h/p cosmos mercury, Isny, Germany, 2.5 m s$^{-1}$), kinematic data of the right leg were captured on the basis of anatomically-referenced reflective markers (greater trochanter, lateral femoral epicondyle and malleolus, fifth metatarsal and calcaneus) by a Vicon motion capture system (250 Hz). A 2 min warm-up and familiarization phase on the treadmill preceded the captured interval. The touchdown of the foot and toe off were defined by the kinematic data as the first and second peak in knee extension, respectively [35]. Ultrasonic images of the soleus were obtained synchronously with a capture frequency of 146 Hz and soleus fascicle length was measured as mentioned above. At least nine steps (11.1 ± 1.5) were analysed for each participant and averaged [8]. Pennation angle was calculated based on the angle between the deeper aponeurosis and the reference fascicle and thickness as distance between both aponeuroses. The corresponding length changes of the soleus muscle belly was calculated as the product of fascicle length and the respective cosine of the pennation angle [36]. Note that this gives not the length of the entire soleus muscle belly but the projection of the instant fascicle length to the plane of the MTU, which can be used to calculate the changes of the belly length. The length change of the soleus MTU was calculated as the product of kinematic data-based ankle angle changes and the individual Achilles tendon lever arm [37], while the initial soleus MTU length was determined at neutral ankle joint angle based on the regression equation provided by Hawkins & Hull [38]. The velocities of MTU, fascicles and muscle belly were calculated as the first derivative of the MTU, fascicle and belly lengths over the time. From these data we calculated the MTU gearing ($V_{MTU}$/$V_{Fascicle}$ [16]), tendon gearing ($V_{MTU}$/$V_{Belly}$ [16]) and belly gearing ($V_{Belly}$/$V_{Fascicle}$ [16,17]), where $V$ is the stance phase-averaged velocity of the soleus MTU, fascicles and muscle belly in absolute (i.e. positive) values. While belly gearing expresses the effects of fascicle rotation, tendon gearing expresses the effects of tendon compliance and MTU gearing is an overall expression of the effects of both components on the fascicle velocity [16].

Surface EMG of soleus was measured by means of the wireless EMG system according to the procedure described above (processing description in electronic supplementary material) and normalized to the maximum processed EMG value obtained from all the individual MVCs (EMG$_{max}$). All parameters were averaged over the same steps as for the muscle fascicle assessment.

## (d) Energetic cost of running

After detaching the ultrasound probe, the participants continued with a 10 min running trial at the same speed (2.5 m s$^{-1}$).

A breath-by-breath cardio pulmonary exercise testing system (MetaLyzer 3B – R2, Cortex Biophysik GmbH, Leipzig, Germany) was used to record the percentage of concentration of both oxygen and carbon dioxide expired and rate of oxygen consumption ($\dot{V}O_2$) and carbon dioxide production ($\dot{V}CO_2$) was calculated as average of the last 3 min. Running economy was expressed in units of energy by

$$\text{energetic cost } = 16.89 \cdot \dot{V}O_2 + 4.84 \cdot \dot{V}CO_2, \qquad (2.1)$$

where the energetic cost is expressed in $[\text{W kg}^{-1}]$ and $\dot{V}O_2$ and $\dot{V}CO_2$ in $[\text{ml s}^{-1}\,\text{kg}^{-1}]$ [3,39].

## (e) Statistics

Differences between soleus MTU and soleus fascicle length changes (absolute and normalized to $L_0$) and velocities as well as between belly gearing and tendon gearing were tested by means of a paired $t$-test for dependent samples. The Pearson correlation coefficient was calculated in order to assess the relationship of the energetic cost of running and the force–velocity potential, force–length–velocity potential and fascicle velocity during the stance phase. As normality was not given for the force–length potential, we used the Spearman correlation coefficient to assess its relationship to the energetic cost. A Pearson correlation coefficient was also used to analyse the relationship of EMG activity (mean and maximum) and force–length–velocity potential. We further conducted a multiple regression analysis to assess the magnitude of the effect of the four independent variables of stance phase-averaged tendon gearing, belly gearing, angular velocity of the ankle joint as well as Achilles tendon lever arm on the absolute soleus fascicle velocity. The statistics were performed using SPSS Statistics (IBM Corp., Version 20.0, Armonk, USA) and the level of significance was set to $\alpha = 0.05$. All values are reported as means and standard deviations.

## 3. Results

The experimentally assessed $L_0$ was on average $41.3 \pm 5.2$ mm and corresponding $F_{max}$ was $2887.1 \pm 724.2$ N. The assessed $V_{max}$ based on the values of $a_{rel} = 0.175$ and $b_{rel} = 1.182\,\text{s}^{-1}$ was $279.0 \pm 34.9$ mm $\text{s}^{-1}$. Achilles tendon lever arm showed an average length of $56.7 \pm 7.4$ mm.

The averaged stance and swing times during running were $304 \pm 23$ ms and $439 \pm 26$ ms, respectively. During the stance phase, the ankle joint showed angles between $17.0 \pm 3.8°$ dorsiflexion and $14.5 \pm 6.0°$ plantarflexion (figure 2), and rotated with an average angular velocity of $164 \pm 12°/\text{s}$. The average activation of soleus normalized to $\text{EMG}_{max}$ throughout the stance phase was $0.32 \pm 0.19\ \text{EMG}_{max}$ and the maximum activation was $0.52 \pm 0.18\ \text{EMG}_{max}$ at $40 \pm 6\%$ of the stance phase (figure 2). While the MTU showed a lengthening–shortening behaviour during the stance phase, the muscle fascicles shortened continuously with significantly less length changes as the MTU ($p < 0.001$; figure 2, table 1). The pennation angle increased coincidentally with fascicle shortening while thickness remained almost unchanged (figure 2, table 1). Operating range (i.e. minimum to maximum) of the fascicles throughout the stance phase covered the top of the ascending limb of the force–length curve ($0.75 \pm 0.09\ L_0$ to $1.01 \pm 0.12\ L_0$; figure 3) with a mean fascicle operating length close to the optimal length (i.e. $0.90 \pm 0.10\ L_0$). Accordingly, the averaged force–length potential of the soleus fascicles was high; (i.e. $0.92 \pm 0.07$; figure 3).

The soleus fascicles operated between $-0.078 \pm 0.045$ $V_{max}$ and $0.322 \pm 0.071\ V_{max}$ with an average velocity of

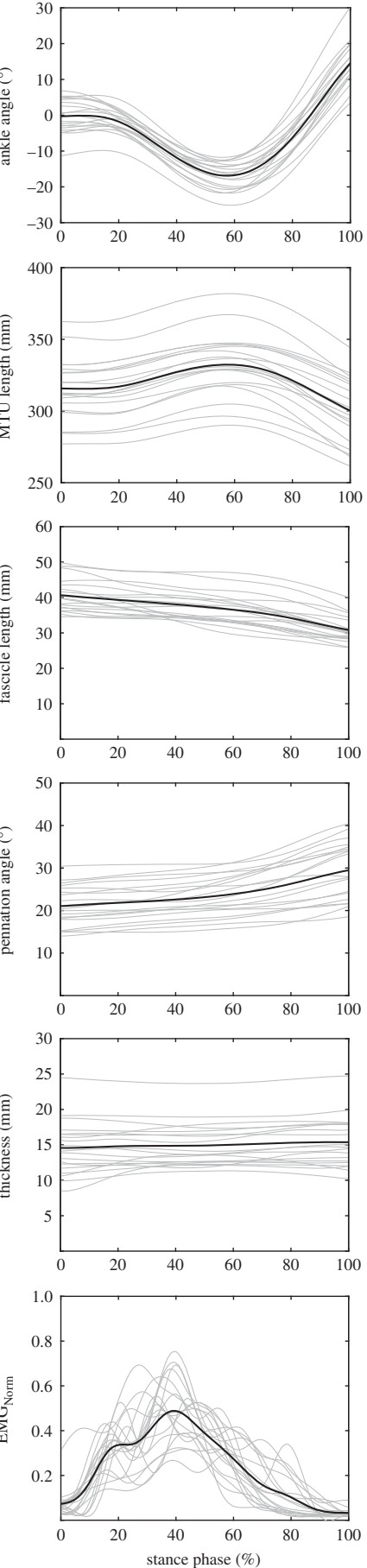

**Figure 2.** Ankle angle, soleus muscle–tendon unit (MTU) length, muscle fascicle length, pennation angle, thickness and electromyographic (EMG) activity (normalized to maximum voluntary isometric contraction) during the stance phase of running ($2.5\ \text{m s}^{-1}$). Individual ($n = 19$) data are shown in thin grey lines and group averages in thick black lines.

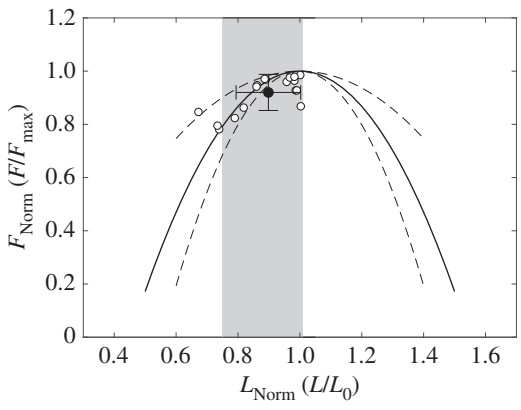
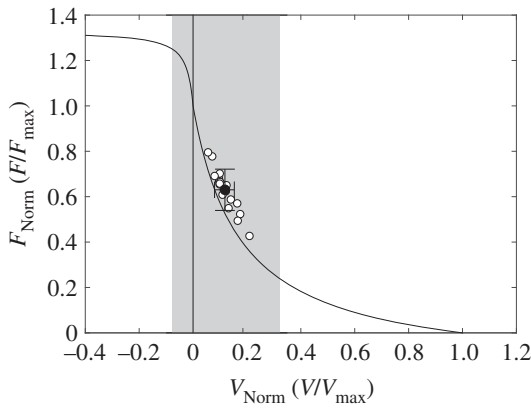

**Figure 3.** Operating length and velocity of soleus muscle fascicles during the stance phase of running mapped onto the averaged normalized force–length and force–velocity curve. White circles indicate the average operating length and velocity of the stance phase of each participant and the black circle the respective group average with the standard deviation of all participants ($n = 19$). The grey shaded areas illustrate the operating range (maximum to minimum) of the operating length and velocity during the stance phase averaged for all participants. Force is normalized to the maximum force during the maximal isometric plantar flexion contractions, fascicle length to the experimentally determined optimal fascicle length and fascicle velocity to the assessed maximum shortening velocity. Dotted lines in the left graph indicate the standard deviation of the individually measured force–length relationships. Note that the data points do not lie on the average curves because the individual force potentials were calculated for each percentage of the stance phase of each step and then averaged step-wise, which makes a difference to the calculation using the overall subject-based average length or velocity due to the non-linearity of the curves.

**Table 1.** Average values (dimension) as well as changes (range) of the ankle joint angle (minus indicates dorsiflexion), soleus muscle–tendon unit (MTU) length and fascicle length (absolute and normalized to optimal length), pennation angle and muscle thickness during the stance phase of running ($n = 19$).

|  | dimension | range |
|---|---|---|
| ankle angle | $-6.1 \pm 3.6°$ | $31.5 \pm 5.2°$ |
| MTU | $321.4 \pm 22.3$ mm | $32.2 \pm 8.2$ mm |
|  |  | $(79.5 \pm 22.9\% L_0)$ |
| fascicles | $36.8 \pm 4.2$ mm | $10.6 \pm 3.0$ mm* |
|  |  | $(25.9 \pm 7.8\% L_0{*})$ |
| pennation angle | $24.0 \pm 5.1°$ | $8.9 \pm 3.1°$ |
| thickness | $15.0 \pm 3.3$ mm | $1.7 \pm 1.0$ mm |

*Statistically significant difference to MTU ($p < 0.05$).

$0.118 \pm 0.039\ V_{max}$ throughout the stance phase (figure 3). The decoupling of MTU and fascicle length trajectories (figure 2) enabled a significantly lower absolute operating velocity of the fascicles ($40.0 \pm 8.2$ mm s$^{-1}$) compared with the MTU ($166.5 \pm 27.7$ mm s$^{-1}$, $p < 0.001$), resulting in a force–velocity potential of $0.63 \pm 0.09$ (figure 3). The achieved total force–length–velocity potential of the soleus muscle during the stance phase of running was $0.58 \pm 0.10$. The calculated velocity gearing ratios were $4.46 \pm 0.98$ for MTU gearing, $4.03 \pm 0.89$ for tendon gearing and $1.11 \pm 0.07$ for belly gearing. The magnitude of tendon gearing was significantly greater ($p < 0.001$) than belly gearing (figure 4).

The energetic cost of running in the investigated velocity of $2.5$ m s$^{-1}$ was in average $10.69 \pm 0.96$ W kg$^{-1}$. An inverse correlation was observed for the energetic cost and the overall force–length–velocity potential of the soleus muscle ($r = -0.520$, $p = 0.022$; figure 5). Energetic cost and the force–velocity potential were also inversely correlated ($r = -0.565$, $p = 0.012$; figure 5), and energetic cost and shortening velocity

were positively correlated ($r = 0.561$, $p = 0.012$), indicative for an association of the economy of running and the operating velocity of the soleus fascicles during running. The force–length potential did not show any significant correlation to the energetic cost ($r_s = -0.076$, $p = 0.759$; figure 5). A significant inverse correlation was also observed for the force–length–velocity potential and the mean ($r = -0.504$, $p = 0.028$) and the maximal EMG activation ($r = -0.525$, $p = 0.021$).

The multiple regression model for the assessment of the fascicle velocity during the stance phase showed a significant explanatory power ($p < 0.001$, $R^2 = 0.928$, adjusted $R^2 = 0.907$) and was expressed by the equation:

Fascicle velocity $= -9.788$ (tendon gearing) $+ 0.716$ (lever arm)
$\qquad - 42.097$ (belly gearing)
$\qquad + 0.209$ (ankle angle velocity) $+ 51.341$.

The four included independent variables were all significant predictors ($p < 0.001$ for tendon gearing, tendon lever arm and belly gearing and $p = 0.002$ for ankle angular velocity). Considering the standardized coefficients of $-1.006$ for tendon gearing, $0.638$ for lever arm, $-0.367$ for belly gearing and $0.310$ for the ankle angular velocity, the model showed that tendon gearing and Achilles tendon lever arm had the greatest effect on the fascicle velocity.

## 4. Discussion

By mapping the operating length and velocity of the human soleus muscle during running onto the individual force–length and force–velocity curves, we investigated the association between the energetic cost of locomotion and the soleus fascicle force–length and force–velocity potential. The findings showed that the soleus fascicles operated close to the optimal length and with moderate continuous shortening during the stance phase. The significant inverse relationship between the energetic cost and the force–velocity potential provides first direct experimental evidence that the fascicle shortening

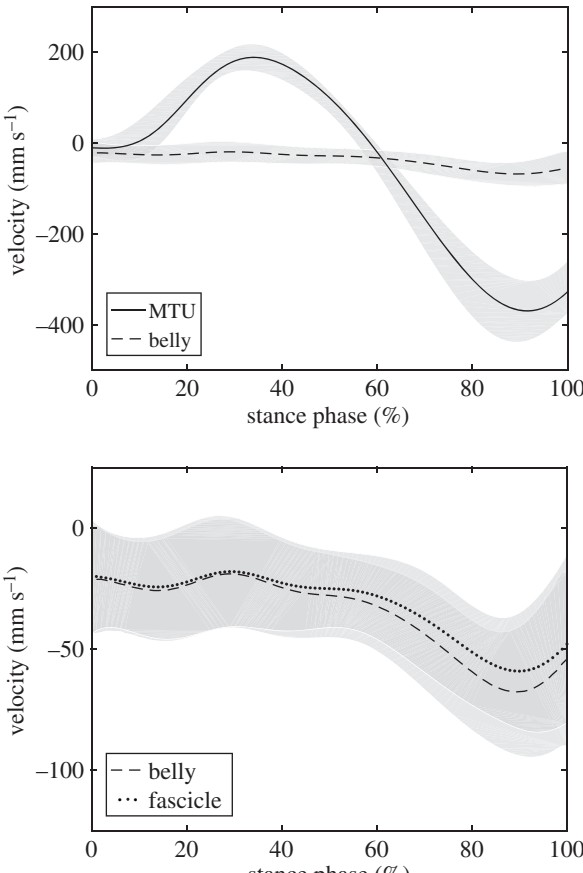

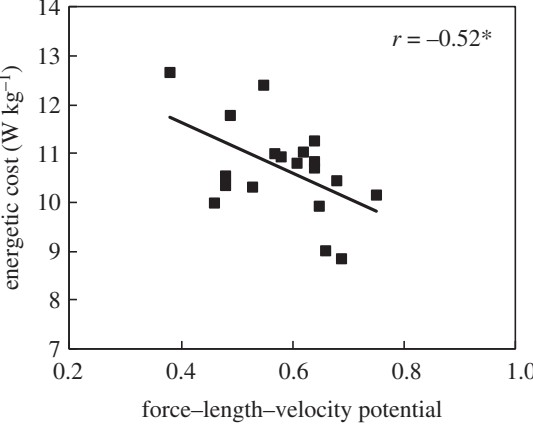

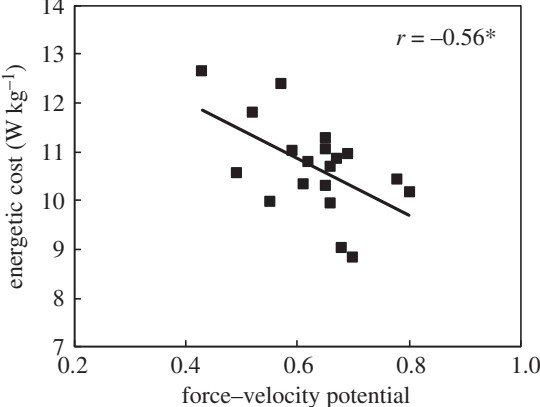

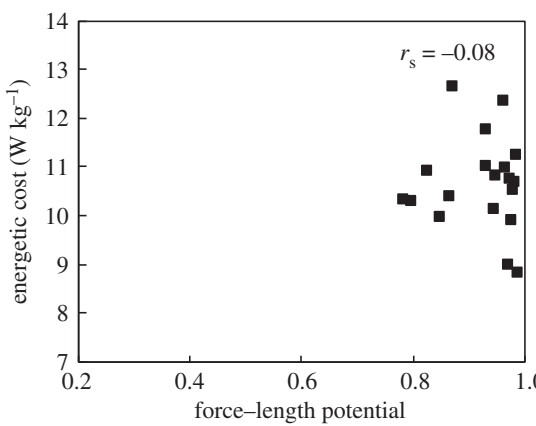

**Figure 4.** Operating velocity of the soleus muscle–tendon unit (MTU) and muscle belly (top) as well as muscle belly and fascicles (bottom) over the stance phase, illustrating the effect of tendon gearing and belly gearing, respectively. Grey shadings indicate the standard deviations ($n = 19$).

**Figure 5.** Association of the force–length–velocity potential, force–velocity potential and force–length potential of the soleus muscle to the energetic cost of running. *Statistically significant correlation ($p < 0.05$).

velocity of the soleus muscle is a determining factor for the economy of human running.

The triceps surae muscle group contributes substantially to the overall energetic cost of running [20]. The soleus is the largest muscle of the triceps surae [40] and the main muscle to lift and accelerate the centre of mass during locomotion [14,15]. During the stance phase of running, the fascicles of the soleus muscle shorten when activated, contributing to the ankle joint mechanical work/power output [41]. The fascicles operated on the steep-rising part of the hyperbolic-shaped force–velocity curve, in average at 11.8% of $V_{max}$, where already small changes in fascicle shortening velocity cause relevant effects on the muscle force–velocity potential. As dictated by the force–velocity relationship [7], an increase in fascicle shortening velocity is accompanied by a decrease in the muscle force potential. The decrease of the muscle force potential requires an upregulation of the muscle activation to maintain the same level of force to support and accelerate the body mass [4,10]. The observed inverse relationship between force–velocity potential and energetic cost confirmed our hypothesis that the soleus fascicle shortening velocity is a key factor for the energetic cost of running. This link may further be supported by the observed inverse correlation of EMG activation and force–length–velocity potential, although it should be considered that active muscle volume cannot be assessed accurately from EMG activity. The fascicles worked in a small range on the upper portion of the ascending limb of the force–length curve with a high force–length potential of 0.92. An operating range on the ascending limb close to $L_0$

(0.75–1.01 $L_0$) was a quite consistent observation in the investigated runners, despite notable differences in the optimal fascicle length ($L_0$ ranging from 33 to 51 mm). In our study, we did not find any relationship between force–length potential and energetic cost of running. However, this does not indicate that the force–length potential is not important for running economy, but rather that the consistently observed high force–length potential explained less of the detected variability in the energetic cost. Besides the favourable high force–length potential for economical force production, operating close to optimal length may additionally preserved from relatively higher energetic cost that can arise when contracting at shorter length. In vitro evidence showed that although force is reduced at shorter sarcomere length, the ATPase rate seems not to differ from the rate at optimal length, indicating

comparably higher cost of contraction at shorter length [42,43]. However, this effect seems more pronounced at very short lengths, a portion of the force–length curve that is probably not covered by the soleus during running (operating range 0.75–1.01 $L_0$). Furthermore, we showed that the soleus shortened continuously during the stance phase of running, which reflects a condition for force depression. However, since a depression of force was shown to be accompanied by a decrease in the ATPase activity [44], force depression would have little or no effect on the energetic cost itself.

During the stance phase, the MTU showed length changes of 80% $L_0$ while the fascicles showed significantly lower length changes (i.e. 26% $L_0$). The regression model provided evidence that the MTU–fascicle decoupling mechanisms of tendon and muscle belly gearing together with the Achilles tendon lever arm and ankle joint angular velocity determine the fascicle velocity. The $R^2$ for the model was 0.928, demonstrating a high goodness-of-fit and a high explanation of variance of the fascicle velocity. The overall MTU gearing ratio of the soleus muscle indicated a 4.5 fold reduction of the fascicle operating velocity during the stance phase of running. The tendon gearing ratio of 4.03 was notably greater than the belly gearing ratio of 1.11, resulting in a higher standardized regression coefficient (−1.006 versus −0.367). The observed gearing ratios indicate that the soleus fascicle velocity during the stance phase of running is mainly governed by the compliance of the series elastic element. The high portion of tendon gearing in the soleus muscle is the consequence of greater length changes of the Achilles tendon and aponeurosis in relation to the muscle belly length changes. The soleus produces mechanical work/energy for the lift and acceleration of the body throughout the entire stance phase. In the first half, where the MTU is elongated, the fascicles actively shorten. This means that a part of the mechanical energy of the human body is transferred to the tendon. Also, in this setting the muscle fascicles produce work under favourable conditions due to the force–length and force–velocity relationships (both potentials in this phase were very high) and save work as strain energy in the tendon. In the second half, the tendon strain energy is returned and at the same time the fascicles produce work by active shortening at a reduced force–velocity potential (fascicle shortening velocity is higher in this phase). The higher shortening velocity is associated with a reduction in the EMG activity and an increase in belly gearing. It has been suggested that increased gearing at fast shortening velocities and lower forces is a mechanism that allows particularly slower type fascicles to be more effective in generating forces [16]. This supports the idea that the observed activation pattern promoted an economical MTU interaction during running.

Belly gearing (or the fascicle rotation component) reduced the shorting velocity of the soleus significantly by 11% in average throughout the stance phase (ratio = 1.11). The main contribution of the fascicle rotation component was in the second half of the stance phase. In situ experiments have shown that belly gearing in pennate muscles is variable with higher ratios during low muscle force to amplify belly shortening at lower fascicle shortening velocity and lower ratios during higher levels of muscle force to facilitate muscle force transmission to the tendon in concentric contractions [17]. In accordance, we found an almost constant belly gearing of ≈1 in the soleus muscle during the first half of the stance phase were activation and consequently force was increased. When the soleus activation level decreased

and the MTU shortened in the second half, the pennation angle increased and enabled a greater contribution of the fascicle rotation component to the reduction of fascicle shortening velocity (maximum belly gearing ratio of 1.18). As proposed by the variable gearing concept, the low fascicle rotation component shown by the soleus muscle during the first half of the stance phase where muscle activation is increased, facilitated the orientation of the line of action of the fascicles to the line of action of the MTU [17].

Our results provide further evidence that the Achilles tendon lever arm and ankle angular excursions during the stance phase were important explanatory factors of the fascicle shortening velocity. The lever arm is an anthropometric characteristic and the results showed that shorter lever arms translated into lower fascicle shortening velocities. The association of the Achilles tendon lever arm and fascicle shortening velocity in the current study provides first direct experimental evidence that shorter lever arms increase the force–velocity potential of the soleus muscle during running. Thus, the reduced fascicle shortening velocity due to a smaller lever arm is—in addition to tendon and belly gearing—a mechanism that improves running economy. Further, the association of the angular velocity of the ankle joint and fascicle shortening velocity during the stance phase shows that greater angular excursions and velocities and in consequence greater length changes of the soleus MTU lead to uneconomical higher fascicle operating velocities.

Although the soleus probably contributes to a great portion of the overall energetic cost during running, other limb muscles that were not considered in the present study are involved. However, the main energy source (positive work) is the ankle joint (41%) [41] and the soleus is the greatest muscle among the main plantar flexors with respect to physiological cross-sectional area (soleus 63%, gastroc. med. 25%, gastroc. lat. 12%) and volume (53%, 31% and 16% [40]). The key role of soleus is further supported by the modelling study of Hamner and Delp (2013), which showed that the soleus is by far the biggest contributor to the vertical acceleration and fore-aft acceleration of the centre of mass [14]. This function is achieved by active shortening, which reduces the force–velocity potential and consequently requires a greater active muscle volume. In contrast, the quadriceps muscle group, the main contributor during early stance, decelerating and supporting body mass [14,15], features more economical fascicle dynamics. Recently, we showed that the fascicles of the vastus lateralis muscle as a representative of the quadriceps muscle group operates with a high force–length (i.e. 0.91) and force–velocity potential (i.e. 0.97) during the stance phase of running [8]. Operating at high force potentials minimizes the cost of this muscle, which is energetically expensive due to its long fascicle length (i.e. $L_0 = 94$ mm [8]), by reducing active muscle volume. This may indicate that the mechanical energy by muscular work required for steady state running is generated by muscles that are metabolically less expensive (i.e. due to shorter fascicle length as the soleus muscle), probably to compensate for the reduction of the force–velocity potential.

To assess the force–velocity potential we used a biologically funded value of $V_{max}$, based on in vitro studies on the human soleus, i.e. 6.77 $L_0$ s$^{-1}$ (279.0 ± 34.9 mm s$^{-1}$). However, during submaximal running in vivo the lower activation level and selective slow fibre type recruitment may affect the actual force–velocity potential of the soleus muscle. To evaluate the effect of the choice of $V_{max}$ on the observed inverse correlation

of force–velocity potential and energetic cost, a sensitivity analysis was conducted by decreasing and increasing $V_{max}$ in 10% intervals and calculating the correlation coefficients, respectively. The results did not show any substantial effects on the associations between force–velocity potential and energetic cost until a value of $V_{max}$ of $<2.0\,L_0\,s^{-1}$ (i.e. $V_{max+30\%}$: $r = -0.577$, $V_{max+20\%}$: $r = -0.574$, $V_{max+10\%}$: $r = -0.570$, $V_{max-10\%}$: $r = -0.559$, $V_{max-20\%}$: $r = -0.552$, $V_{max-30\%}$: $r = -0.544$, $V_{max-40\%}$: $r = -0.534$, $V_{max-50\%}$: $r = -0.522$, $V_{max-60\%}$: $r = -0.506$, $V_{max-70\%}$: $r = -0.484$; $p < 0.05$), which confirms and strengthens the overserved association. Furthermore, we assessed the force–length curve during maximal isometric contractions and used it to calculate the force–length potential of the soleus muscle during running at submaximal activation. There is evidence from *in vitro* studies that the force–length curve depends on muscle activation [45,46]. However, in a recent *in vivo* study by Fontana & Herzog [47] on the human vastus lateralis muscle, a rightward shift of optimal length with submaximal activation was not observed when force was normalized to the maximum EMG signal. The authors suggested that the shift in optimal length phenomenon might be related to the *in vitro* testing set-up (e.g. non-physiological stimulation frequency range or $Ca^{2+}$ concentrations). The discrepancy of the *in vitro* and *in vivo* evidence clearly warrants future investigation to elucidate the shifting length phenomenon in the context of *in vivo* submaximal locomotion. Given the current human *in vivo* evidence [47], we can argue that mapping the submaximal fascicle operating length onto the force–length curve in the present *in vivo* study should not affect the findings.

In the present study we focused on the understanding of the contribution of the force–length and force–velocity potential to the energetic cost of running and we showed that the force–velocity potential is inversely related to the energetic cost, explaining about one-third of its variance. We argue that an increase of active muscle volume due to the decreased force–velocity potential would increase the energetic cost of running. However, it must be acknowledged that the energetic cost of muscle contraction is complex and multifactorial.

Independent of active muscle volume, in higher shortening velocities the rate of cross-bridge cycling is increased and as a consequence so is the consumed energy. In our study, shortening velocities of the soleus muscle were on average $0.118\,V_{max}$ throughout the stance phase, a range where the rate of ATP hydrolysis shows a steep increase [48]. Furthermore, in submaximal intensity contractions as during our investigated running velocity selective slow fibre type activation might decrease the energetic cost by reducing the contribution of energetically more expensive fast twitch fibres.

## 5. Conclusion

In conclusion, this study provides for the first time experimental evidence that the energetic cost of running is related to the force–length–velocity potential of the soleus muscle with lower shortening velocities of the fascicles as the main influencing factor (i.e. higher force–velocity potential). The main mechanism for the underlying reduction of the fascicle shortening velocity during the stance phase was gearing within the MTU, particularly greater tendon gearing, a shorter Achilles tendon lever arm as well as, to a minor extent, a lower ankle angular velocity.

Ethics. The ethics committee of the Humboldt-Universität zu Berlin approved the study and the participants gave written informed consent in accordance with the Declaration of Helsinki.

Data accessibility. The datasets generated and analysed during the current study are available at https://dx.doi.org/10.6084/m9.figshare.10119320.v1.

Authors' contributions. S.B., F.M. and A.A. designed research; S.B., F.M. and A.S. performed research; S.B. analysed data; S.B. and A.A. drafted the manuscript and F.M. and A.S. made important intellectual contributions during revision.

Competing interests. We declare we have no competing interests.

Funding. Funding for this research was supplied by the German Federal Institute of Sport Science (grant no. ZMVI14-070604/17-18).

Acknowledgements. We acknowledge the support by Theresa Domroes, Antonis Ekizos and Arno Schroll for data analysis.

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
