## [Reviewer comments · Proceedings of the Royal Society B: Biological Sciences]

Review History

RSPB-2019-1961.R0 (Original submission)

Review form: Reviewer 1

Recommendation

Accept with minor revision (please list in comments)

Scientific importance: Is the manuscript an original and important contribution to its field?

Excellent

General interest: Is the paper of sufficient general interest?

Excellent

Quality of the paper: Is the overall quality of the paper suitable?

Excellent

Is the length of the paper justified?

Yes

Should the paper be seen by a specialist statistical reviewer?

No

Do you have any concerns about statistical analyses in this paper? If so, please specify them explicitly in your report.

No

It is a condition of publication that authors make their supporting data, code and materials available - either as supplementary material or hosted in an external repository. Please rate, if applicable, the supporting data on the following criteria.

Is it accessible?

N/A

Is it clear?

N/A

Is it adequate?

N/A

Do you have any ethical concerns with this paper?

No

Comments to the Author

This study tests the fascicle length and shortening velocity (imaged using ultrasound) of the soleus during running at a moderate speed in 19 participants. The force-length properties are also measured in a series of isometric contractions on a dynamometer, and the force-velocity properties of the muscle are estimated from the literature. The soleus fascicles are assessed by how close they are to optimal length (force length potential) and how slow they shorten (force velocity potential), and these are compared to the actual cost of running as measured using expired gas analysis from these same participants. It is shown that the fascicles have a high but constant force length potential, and a high force velocity potential: this force velocity potential varies between participants and was related to the running economy. The fascicle velocity was further related (via multi-regression analysis) to the tendon gearing, Achilles tendon moment arm, belly gearing and ROM in these participants.

This is an interesting and timely question, and it is posed well. A thorough set of experimental data are collected for the analysis, and the data appear to have been collected very well. The data show convincing relations and support the hypothesis. The paper is written clearly, and the figures are appropriate.

Major comments:

1. Choice of maximum shortening velocity. The major finding is that the running economy is sensitive to the force-velocity potential: the absolute values of these depend on the V_{max} and curvature of the force-velocity relation. It is not clear where the value of $V_{max} = 11.72 \text{ s}^{-1}$ comes from. I have followed the cited papers back through several layers of citations, and cannot find definitive justification for these values beyond a statement that V_{max} is typically 10-12 s^{-1} for modelling studies: this in itself does not justify the value.

Note that actual measurements of V_{max} in humans are rare. Reports of V_{max} of 2 and 6 s^{-1} have been made for isolated bundles of slow and fast human muscle fibres, respectively (Faulkner et al. 1986), and it has been argued that these values could be less than 8 s^{-1} and greater than 14 s^{-1} , respectively (Epstein & Herzog 1998) from intact human experiments. Higher values are used for musculoskeletal simulations in part to overcome the tendency of simulations to under-predict muscle forces, but this is not a scientific justification for inflating these values.

This submitted manuscript would benefit from a clearer description and rationale for the force-velocity parameters chosen. However, I note (as also discussed on line 370) that the actual choice of value has little effect on the conclusions in the study.

2. Multiple regression. The variables used in the multiple regression all pertain to the length change of the soleus muscle fascicles. However, fascicle velocity is a function of change in length and the time taken for this change. Thus, the time taken for the muscle shortening should be considered. Was this constant across participants? Was this considered for the multiple regression (and removed later because it had no effect - this would be a good approach but should be reported)?

3. Experimental details. Additional details should be presented in the Methods section for how the EMG magnitude is quantified, and how fascicles are identified in the ultrasound images.

Minor comment.

1. State in the text whether errors are standard deviations or standard errors of the mean.

Review form: Reviewer 2 (Natalie Holt)

Recommendation

Major revision is needed (please make suggestions in comments)

Scientific importance: Is the manuscript an original and important contribution to its field?

Acceptable

General interest: Is the paper of sufficient general interest?

Acceptable

Quality of the paper: Is the overall quality of the paper suitable?

Good

Is the length of the paper justified?

Yes

Should the paper be seen by a specialist statistical reviewer?

No

Do you have any concerns about statistical analyses in this paper? If so, please specify them explicitly in your report.

No

It is a condition of publication that authors make their supporting data, code and materials available - either as supplementary material or hosted in an external repository. Please rate, if applicable, the supporting data on the following criteria.

Is it accessible?

N/A

Is it clear?

N/A

Is it adequate?

N/A

Do you have any ethical concerns with this paper?

No

Comments to the Author

See attached file. (See Appendix A)

Decision letter (RSPB-2019-1961.R0)

23-Sep-2019

Dear Dr Bohm:

I am writing to inform you that your manuscript RSPB-2019-1961 entitled "The force-length and force-velocity potential of the human soleus muscle is related to the energetic cost of running" has, in its current form, been rejected for publication in Proceedings B.

This action has been taken on the advice of referees, who have recommended that substantial revisions are necessary. With this in mind we would be happy to consider a resubmission, provided the comments of the referees are fully addressed. However please note that this is not a provisional acceptance. While both reviewers see value to the study, both have concerns about some of the assumptions and other aspects of the methods/interpretation. They and the Associate Editor need to be won over more if this study is to be accepted.

In your revision process, please take a second look at how open your science is; our policy is that all data involved with the study should be made openly accessible-- see: <https://royalsociety.org/journals/ethics-policies/data-sharing-mining/>

Insufficient sharing of data can delay or even cause rejection of a paper.

Sincerely,
Professor John Hutchinson, Editor
mailto: proceedingsb@royalsociety.org

Associate Editor
Board Member: 1
Comments to Author:
Associate Editor: Douglas L Altshuler

The authors have performed an integrative study of muscle physiology and energetics during running. The authors and I agree that the methods are sound and creative, and the results are clear and interesting. One of the referees expressed some concern that the study may be too narrow in scope for the general science audience of ProcB. I would encourage the authors to revise their manuscript, and it would be helpful to see how this concern about breadth could be addressed.

Reviewer(s)' Comments to Author:

Referee: 1

Comments to the Author(s)

This study tests the fascicle length and shortening velocity (imaged using ultrasound) of the soleus during running at a moderate speed in 19 participants. The force-length properties are also measured in a series of isometric contractions on a dynamometer, and the force-velocity properties of the muscle are estimated from the literature. The soleus fascicles are assessed by how close they are to optimal length (force length potential) and how slow they shorten (force velocity potential), and these are compared to the actual cost of running as measured using expired gas analysis from these same participants. It is shown that the fascicles have a high but constant force length potential, and a high force velocity potential: this force velocity potential varies between participants and was related to the running economy. The fascicle velocity was further related (via multi-regression analysis) to the tendon gearing, Achilles tendon moment arm, belly gearing and ROM in these participants.

This is an interesting and timely question, and it is posed well. A thorough set of experimental data are collected for the analysis, and the data appear to have been collected very well. The data show convincing relations and support the hypothesis. The paper is written clearly, and the figures are appropriate.

Major comments:

1. Choice of maximum shortening velocity. The major finding is that the running economy is sensitive to the force-velocity potential: the absolute values of these depend on the V_{max} and curvature of the force-velocity relation. It is not clear where the value of $V_{max} = 11.72 \text{ s}^{-1}$ comes from. I have followed the cited papers back through several layers of citations, and cannot find definitive justification for these values beyond a statement that V_{max} is typically 10-12 s^{-1} for modelling studies: this in itself does not justify the value.

Note that actual measurements of V_{max} in humans are rare. Reports of V_{max} of 2 and 6 s^{-1} have been made for isolated bundles of slow and fast human muscle fibres, respectively (Faulkner et al. 1986), and it has been argued that these values could be less than 8 s^{-1} and greater than 14 s^{-1} , respectively (Epstein & Herzog 1998) from intact human experiments. Higher values are used for

musculoskeletal simulations in part to overcome the tendency of simulations to under-predict muscle forces, but this is not a scientific justification for inflating these values.

This submitted manuscript would benefit from a clearer description and rationale for the force-velocity parameters chosen. However, I note (as also discussed on line 370) that the actual choice of value has little effect on the conclusions in the study.

2. Multiple regression. The variables used in the multiple regression all pertain to the length change of the soleus muscle fascicles. However, fascicle velocity is a function of change in length and the time taken for this change. Thus, the time taken for the muscle shortening should be considered. Was this constant across participants? Was this considered for the multiple regression (and removed later because it had no effect – this would be a good approach but should be reported)?

3. Experimental details. Additional details should be presented in the Methods section for how the EMG magnitude is quantified, and how fascicles are identified in the ultrasound images.

Minor comment.

1. State in the text whether errors are standard deviations or standard errors of the mean.

Referee: 2

Comments to the Author(s)

See attached file

Author's Response to Decision Letter for (RSPB-2019-1961.R0)

See Appendix B.

RSPB-2019-2560.R0

Review form: Reviewer 1

Recommendation

Accept with minor revision (please list in comments)

Scientific importance: Is the manuscript an original and important contribution to its field?

Excellent

General interest: Is the paper of sufficient general interest?

Excellent

Quality of the paper: Is the overall quality of the paper suitable?

Excellent

Is the length of the paper justified?

Yes

Should the paper be seen by a specialist statistical reviewer?

No

Do you have any concerns about statistical analyses in this paper? If so, please specify them explicitly in your report.

No

It is a condition of publication that authors make their supporting data, code and materials available - either as supplementary material or hosted in an external repository. Please rate, if applicable, the supporting data on the following criteria.

Is it accessible?

N/A

Is it clear?

N/A

Is it adequate?

N/A

Do you have any ethical concerns with this paper?

No

Comments to the Author

The authors have addressed all my previous concerns in a careful manner.

There remains one further comment that they may choose to consider for the manuscript, and it still concerns the choice of V_{max} . I appreciate the further analysis that the authors have attempted, to provide a value of V_{max} for the Soleus. However, it should be noted that the running velocity of 2.5 m/s is not all that fast, and indeed the EMG averages less than 50%. As such, it is likely that the fastest muscle fibres will not have been recruited, and hence the weighted mean taken for V_{max} may thus be an overestimate. Coupled to this, with more than half of the muscle inactive, the actual V_{max} may be less than its constituent fibres (for additional reasons: Holt et al. Proc Roy Soc B 2014). If the V_{max} for the Soleus were less than the estimated 6.77 L/s for this experimental situation, then it is likely that the actual spread of Force-velocity potentials would be larger than shown in Fig. 3. It is thus worth considering that you have actually resulted with a conservative evaluation of the importance of the force-velocity potential.

Review form: Reviewer 2 (Natalie Holt)

Recommendation

Accept with minor revision (please list in comments)

Scientific importance: Is the manuscript an original and important contribution to its field?

Good

General interest: Is the paper of sufficient general interest?

Good

Quality of the paper: Is the overall quality of the paper suitable?

Good

Is the length of the paper justified?

Yes

Should the paper be seen by a specialist statistical reviewer?

No

Do you have any concerns about statistical analyses in this paper? If so, please specify them explicitly in your report.

No

It is a condition of publication that authors make their supporting data, code and materials available - either as supplementary material or hosted in an external repository. Please rate, if applicable, the supporting data on the following criteria.

Is it accessible?

N/A

Is it clear?

N/A

Is it adequate?

N/A

Do you have any ethical concerns with this paper?

No

Comments to the Author

See attached file. (See Appendix C)

Decision letter (RSPB-2019-2560.R0)

20-Nov-2019

Dear Dr Bohm

I am pleased to inform you that your manuscript RSPB-2019-2560 entitled "The force-length and force-velocity potential of the human soleus muscle is related to the energetic cost of running" has been accepted for publication in Proceedings B. Congratulations!!

The referee(s) have recommended publication, but also suggest some minor revisions to your manuscript. Therefore, I invite you to respond to the referee(s)' comments and revise your manuscript. Because the schedule for publication is very tight, it is a condition of publication that you submit the revised version of your manuscript within 7 days. If you do not think you will be able to meet this date please let us know.

To revise your manuscript, log into <https://mc.manuscriptcentral.com/prsb> and enter your Author Centre, where you will find your manuscript title listed under "Manuscripts with

Decisions." Under "Actions," click on "Create a Revision." Your manuscript number has been appended to denote a revision. You will be unable to make your revisions on the originally submitted version of the manuscript. Instead, revise your manuscript and upload a new version through your Author Centre.

NB. From April 1 2013, peer reviewed articles based on research funded wholly or partly by RCUK must include, if applicable, a statement on how the underlying research materials – such

as data, samples or models – can be accessed. This statement should be included in the data accessibility section.

[http://datadryad.org/submit?journalID=RSPB&manu=\(Document not available\)](http://datadryad.org/submit?journalID=RSPB&manu=(Document%20not%20available)) which will take you to your unique entry in the Dryad repository. If you have already submitted your data to dryad you can make any necessary revisions to your dataset by following the above link. Please see <https://royalsociety.org/journals/ethics-policies/data-sharing-mining/> for more details.

Sincerely,
Professor John Hutchinson, Editor
mailto: proceedingsb@royalsociety.org

Associate Editor
Board Member
Comments to Author:
Associate Editor: Doug Altshuler

The authors have done a good job addressing the reviewer concerns. A few issues remain, and I agree with the reviewers that consideration of these final points would further strengthen the manuscript.

Reviewer(s)' Comments to Author:
Referee: 2

Comments to the Author(s).
See attached file

Referee: 1

Comments to the Author(s).
The authors have addressed all my previous concerns in a careful manner.

There remains one further comment that they may choose to consider for the manuscript, and it still concerns the choice of V_{max} . I appreciate the further analysis that the authors have attempted, to provide a value of V_{max} for the Soleus. However, it should be noted that the running velocity of 2.5 m/s is not all that fast, and indeed the EMG averages less than 50%. As such, it is likely that the fastest muscle fibres will not have been recruited, and hence the weighted mean taken for V_{max} may thus be an overestimate. Coupled to this, with more than half of the muscle inactive, the actual V_{max} may be less than its constituent fibres (for additional reasons: Holt et al. Proc Roy Soc B 2014). If the V_{max} for the Soleus were less than the estimated 6.77 L/s for this experimental situation, then it is likely that the actual spread of Force-velocity

potentials would be larger than shown in Fig. 3. It is thus worth considering that you have actually resulted with a conservative evaluation of the importance of the force-velocity potential.

Author's Response to Decision Letter for (RSPB-2019-2560.R0)

See Appendix D.

Decision letter (RSPB-2019-2560.R1)

26-Nov-2019

Dear Dr Bohm

I am pleased to inform you that your manuscript entitled "The force-length and force-velocity potential of the human soleus muscle is related to the energetic cost of running" has been accepted for publication in Proceedings B.

Open Access

Paper charges

All supplementary materials accompanying an accepted article will be treated as in their final form. They will be published alongside the paper on the journal website and posted on the online

figshare repository. Files on figshare will be made available approximately one week before the accompanying article so that the supplementary material can be attributed a unique DOI.

Sincerely,
Proceedings B
<mailto:proceedingsb@royalsociety.org>

Appendix A

In this study, the authors explore the effect of muscle force-length and force-velocity conditions on the energetic cost of running. In addition, they explore the determinants of muscle fiber length change. I find this to be an extensive and well collected data set that uses *in vivo* determination of force-length and force-velocity relationships, and application of these to muscle function during running to address these questions.

There appear to me to be a few major limitations of the study. These should be addressed throughout.

- 1) Organismal energy consumption is measured, but only the length and velocity profile of soleus. This prevents the authors from drawing the more interesting conclusion that muscle shortening velocity is a determinant of energy consumption. This is acknowledged and somewhat addressed in the discussion, but remains a major limitation to the study.
- 2) The strict adherence to force-length and force-velocity relationships as defining features of muscle performance seems somewhat outdated given a wealth of literature showing that these relationships do not hold under conditions relevant to locomotion (i.e. history dependence, activation-dependent changes). These advances do not negate this study, however, it would be a more accurate representation of the field to acknowledge that they exist, and present this study as a means to investigate the importance of these relationships.
- 3) This paper is fundamentally concerned with the effect of contractile conditions on muscle energy consumption. However, there is very little discussion of why length and velocity might affect energy consumption beyond required activation, despite a wealth of evidence on this i.e. how the cost per unit force varies across the force-length relationship in isolated muscle. In addition, it may be worth considering findings such as the effect of contractile history on cost (Joumaa et al., 2013), and the complexity of the cost of work (Holt et al., 2014; Curtin et al., 2019) in a more comprehensive discussion of *in vivo* muscle energetics.

Specific comments

Lines 50-51 – The ongoing debating between the cost of force and the cost of work as determinants of organismal cost should be acknowledged here. This could then also lead to a more nuanced discussion of factors dictating muscle energetics beyond simply level of activation.

Lines 123-124 and 185-187 – It is relatively unclear to me how the force-velocity relationship was determined here. It appears as though force and velocity were determined as fibers shortened against the tendon? Can the authors make this clearer, better define where in the contraction force and velocity were determined, and comment on how this might affect findings compared to a more standard isotonic or isovelocity protocol.

Line 197-198 – The meaning of this is unclear to me. This description of touchdown and toe-off should be reworded for clarification

The results section is relatively dense. The authors may wish to consider moving some of the findings less critical to addressing their question to a table, to improve readability.

Line 299-300 – The assertion that the triceps surae consumes 40% of the cost during running is crucial to the argument of this paper. Yet it is not clear how this value is arrived at from the Fletcher and MacIntosh paper cited (the paper seems to give a large range of values for muscle energy consumption and not to relate this to organismal cost), and how reliable the output of their simple model is for this purpose. Could the authors give a little more detail on this (in the manuscript if of sufficient interest, or simply here). It may also be useful to combine this 40% estimate with the relative size of soleus to give a better representation of its likely contribution to energy consumption, considering fiber type as soleus is likely cheaper than gastrocs (Barclay, 1993).

Lines 304-309 - The authors make a good case for why small changes in velocity would require an increase in activation and therefore cost. This effect should be seen in EMG recordings. It would seem that the argument could be strengthened by showing this as it would provide a more causal link between the change in muscle level function and organismal level cost.

Line 344-345 – There seems to be some discrepancy regarding activation in here. The implication seems to be that muscle activation is higher in early stance to enable the tendon to be stretched, and then recoil to slow shortening velocity in the later part of stance. Yet a central claim of the paper is that cost is lower when shortening velocity is lower, due to a lower requirement for activation. It seems like the variation in required activation could balance out over the course of a stance phase? Could it be clarified as to why the early increase in activation to enable tendon stretch doesn't seem to be costly in the way that the latter reduction is deemed to be cheap?

Line 345 – typo “were”?

Line 375 – The study doesn't seem to show that energy consumption is related to the force-length-velocity potential, but rather just the force-velocity potential.

Appendix B

Referee: 1

Comment: This study tests the fascicle length and shortening velocity (imaged using ultrasound) of the soleus during running at a moderate speed in 19 participants. The force-length properties are also measured in a series of isometric contractions on a dynamometer, and the force-velocity properties of the muscle are estimated from the literature. The soleus fascicles are assessed by how close they are to optimal length (force length potential) and how slow they shorten (force velocity potential), and these are compared to the actual cost of running as measured using expired gas analysis from these same participants. It is shown that the fascicles have a high but constant force length potential, and a high force velocity potential: this force velocity potential varies between participants and was related to the running economy. The fascicle velocity was further related (via multi-regression analysis) to the tendon gearing, Achilles tendon moment arm, belly gearing and ROM in these participants.

This is an interesting and timely question, and it is posed well. A thorough set of experimental data are collected for the analysis, and the data appear to have been collected very well. The data show convincing relations and support the hypothesis. The paper is written clearly, and the figures are appropriate.

Response: *Thank you for your valuable comments. All changes are underlined in the revised version of the manuscript and the references cited in the responses can be found at the end of the document. Please note that some parts of the methods are now presented in the electronic supplementary material due to length restrictions of the journal.*

Major comments:

Comment: 1. Choice of maximum shortening velocity. The major finding is that the running economy is sensitive to the force-velocity potential: the absolute values of these depend on the V_{max} and curvature of the force-velocity relation. It is not clear where the value of $V_{max} = 11.72 \text{ s}^{-1}$ comes from. I have followed the cited papers back through several layers of citations, and cannot find definitive justification for these values beyond a statement that V_{max} is typically $10\text{-}12 \text{ s}^{-1}$ for modelling studies: this in itself does not justify the value. Note that actual measurements of V_{max} in humans are rare. Reports of V_{max} of 2 and 6 s^{-1} have been made for isolated bundles of slow and fast human muscle fibres, respectively (Faulkner et al. 1986), and it has been argued that these values could be less than 8 s^{-1} and greater than 14 s^{-1} , respectively (Epstein & Herzog 1998) from intact human experiments. Higher values are used for musculoskeletal simulations in part to overcome the tendency of simulations to under-predict muscle forces, but this is not a scientific justification for inflating these values. This submitted manuscript would benefit from a clearer description and rationale for the force-velocity parameters chosen. However, I note (as also discussed on line 370) that the actual choice of value has little effect on the conclusions in the study.

Response: *Thank you for this comment. Indeed, experimental assessed values of V_{max} for human muscles in vivo are very rare and for the soleus not reported so far to our knowledge. This was the reason why we based our calculations on recommendations for modelling approaches [1,2]. When extending our sensitivity analysis about the effect of the magnitude of V_{max} on the correlation of the*

force-velocity potential and energetic cost, we found that the correlation remained statistically significant ($p < 0.05$) for V_{max} values higher than 3.0 L_0/s . Therefore, the observed association of force-velocity potential and energetic cost seems quite strong. Furthermore, we reported a direct correlation between the operating velocity and the energetic cost ($r = 0.561$ $p = 0.012$). Since this association is independent of the choice of V_{max} , we can be confident about the general study findings.

Biological support for the choice of V_{max} could be derived indirectly when referring to *in vitro* studies on the human soleus muscle. Luden et al., (2008) showed V_{max} values for MHC I type fibers of 0.77 L_0/s and 2.91 L_0/s for MHC IIA type fibers measured at 15°C [3]. Considering the temperature coefficient provided by Ranatunga et al., (1984) [4] it can be predicted that V_{max} would increase to 4.4 L_0/s for MHC I type fibers and to 16.8 L_0/s for MHC IIA type fibers under physiological temperature conditions (37 °C). The fiber type distribution in the human soleus muscle can be estimated from literature reports, i.e. Johnson et al., 1973: type 1 fibers 87,7%, type 2 fibers (a and b) 12,3% (average of surface and deep fiber location) [5]; Larsson and Moss, 1993: type 1 89%, type 2A 11% [6]; Edgerton et al., 1975: slow twitch 70%, fast twitch 30% [7]; Luden et al., 2008: 74% MHC I, 20% MHC IIA (norm to 100%) [3]). Using an average of those reported distribution values (type 1: 81%, type 2: 19%), V_{max} for soleus under physiological temperature can be calculated as 6.77 L_0/s . The broad literature basis for the average fiber type distribution was also used to update a_{rel} to 0.175 (i.e. $0.1 + 0.4FT$, where FT is the fast twitch fiber type percentage [8,9]) and accordingly b_{rel} to 1.182 ($a_{rel} * V_{max}$ [10]).

We addressed the comment of the reviewer and recalculated the respective values (force-velocity potential and normalized velocities and their ranges) using the updated V_{max} of 6.77 L_0/s , a_{rel} and b_{rel} in the revised manuscript. Again, this did not change any of the statistical correlation outcomes.

We added the following information to the revised manuscript (page: 4, line: 155):

“Furthermore, we assessed the force-velocity relationship of soleus using the classical Hill equation [11], the muscle-specific maximum fascicle shortening velocity (V_{max}) and constants of a_{rel} and b_{rel} . V_{max} was derived from the study of Luden et al. (2008), which showed V_{max} values for type 1 fibers of 0.77 L_0/s and 2.91 L_0/s for type 2 fibers of the human soleus muscle measured *in vitro* at 15°C [3]. Considering the temperature coefficient [4], V_{max} can be predicted as 4.4 L_0/s for type 1 fibers and 16.8 L_0/s for type 2 fibers under physiological temperature conditions (37 °C). Using an average fiber type distribution (type 1 fibers: 81%, type 2: 19%) of the human soleus muscle reported in literature [3,5–7], V_{max} can be calculated as 6.77 L_0/s . a_{rel} was calculated as $0.1 + 0.4FT$, where FT is the fast twitch fiber type percentage (see above), which then equals to 0.175 [8,9]. The product of a_{rel} and V_{max} then gives b_{rel} as 1.182 [10]. After rearrangement of the Hill equation and extension to the eccentric component, the operating velocity normalized to V_{max} can be used to calculate the individual force potential according to the force-velocity relationship.”

Comment: 2. Multiple regression. The variables used in the multiple regression all pertain to the length change of the soleus muscle fascicles. However, fascicle velocity is a function of change in length and the time taken for this change. Thus, the time taken for the muscle shortening should be considered. Was this constant across participants? Was this considered for the multiple regression (and removed later because it had no effect – this would be a good approach but should be reported)?

Response: Thanks for this comment. The time for the muscle shortening (i.e. stance time) showed some variability among the participants as to be expected, i.e. mean 304 ms, SD 23.1 ms, maximum 362 ms, minimum 270 ms.

With respect to the four variables (tendon gearing, belly gearing, tendon lever arm and ankle angle range) and the effect of time, we reasoned that the tendon and belly gearing are ratios between velocities, and are thereby independent of time (i.e. tendon gearing: V_{MTU}/V_{Belly} and belly gearing: $V_{Belly}/V_{Fascicle}$, where V is the stance phase-averaged velocity). The tendon lever arm is a quantity that is also independent from time, while only the ankle angle range is time-dependent. We now calculated the stance phase-averaged absolute ankle angle velocity and rebuilt the regression model, which is now expressed by the updated equation:

$$\text{Fascicle velocity} = -9.788 (\text{tendon gearing}) + 0.716 (\text{lever arm}) - 42.097 (\text{belly gearing}) + 0.209 (\text{mean ankle angle velocity}) + 51.341$$

The model remained significant ($p < 0.001$, $R^2 = 0.928$, adjusted $R^2 = 0.907$) and so did the four independent variables ($p < 0.001$ for tendon gearing, tendon lever arm and belly gearing, and $p = 0.002$ for ankle angle velocity). The standardized coefficients changed slightly to -1.006 for tendon gearing, 0.638 for lever arm, -0.367 for belly gearing and 0.310 for the ankle angle velocity.

We changed this part in the revised manuscript ((page: 6, line: 262; page: 7, line: 315).

Comment: 3. Experimental details. Additional details should be presented in the Methods section for how the EMG magnitude is quantified, and how fascicles are identified in the ultrasound images.

Response: We amended the information on the EMG assessment during running in the revised manuscript as supplementary material ("EMG processing"). Note that the low-pass filter cut-off frequency was changed from 6 Hz to 20 Hz.

"Raw EMG signals (running and MVC) were processed by a fourth-order high-pass Butterworth zero-phase filter with a 50 Hz cut-off frequency then a full-wave rectification and a low-pass zero-phase filter with a 20 Hz cut-off frequency for creating a linear envelope of the signal [12,13]."

We added some more information on the fascicle identification in the revised manuscript as supplementary material ("Fascicle length determination from the ultrasound images"):

"The procedure included an approximation of the deeper and upper aponeurosis by a best linear fit through three manually placed and frame-by-frame adjusted marks. By means of the bwtraceboundary function of the Matlab Image Processing toolbox the algorithm then identified the shape and orientation of image brightness features between both aponeuroses in each frame, which are indicative for the hyperechoic perimysial connective tissue parts aligned with the muscle fascicles (fig. 1A). The feature identification criteria were set to: minimal length of 23 pixels (i.e. 0.4 cm, from the bottom left to the top right), area to length ratio of 8.5, angle between feature and deeper aponeurosis between 10° and 70° and 80% of the pixels on a line between the start and end point of a feature had to be white [14]. Every frame was visually controlled for adequate feature placement and manually corrected if necessary. Based on the identified features, a linear averaged reference fascicle was calculated (fig. 1A). Reliability of the tracking approach was confirmed and reported in two previous studies [14,15]."

Minor comment.

Comment: 1. State in the text whether errors are standard deviations or standard errors of the mean.

Response: *We added the information that standard deviations are presented in the revised manuscript (page: 5, line: 221).*

Referee: 2

Comment: In this study, the authors explore the effect of muscle force-length and force-velocity conditions on the energetic cost of running. In addition, they explore the determinants of muscle fiber length change. I find this to be an extensive and well collected data set that uses in vivo determination of force-length and force-velocity relationships, and application of these to muscle function during running to address these questions. There appear to me to be a few major limitations of the study. These should be addressed throughout.

Response: *Thank you for your valuable comments. All changes are underlined in the revised version of the manuscript and the references cited in the responses can be found at the end of the document. Please note that some parts of the methods are now presented in the electronic supplementary material due to length restrictions of the journal.*

Comment: 1) Organismal energy consumption is measured, but only the length and velocity profile of soleus. This prevents the authors from drawing the more interesting conclusion that muscle shortening velocity is a determinant of energy consumption. This is acknowledged and somewhat addressed in the discussion, but remains a major limitation to the study.

Response: *We agree with the reviewers comment that although the soleus contributes to a great portion of the overall energetic costs during running [16] other limb muscles are involved that were not considered in the present study and this remains a limitation. However, the main energy source (positive work) is the ankle joint (41%) while the contribution of the knee and hip joint is comparably lower during running [17]. The soleus is the greatest muscle among the main plantar flexors with respect to physiological cross-sectional area and volume (soleus: 131 cm² and 477 cm³, gastrocnemius medialis: 51 cm² and 285 cm³, gastrocnemius lateralis: 24 cm² and 146 cm³ [18]), giving this muscle a key role. Using a modeling approach, Hamner and Delp (2013) showed that the contribution of soleus to the vertical acceleration of the center of mass at the same velocity as in our study (i.e. 2.5 m/s) is remarkably higher than those of the other lower limb muscles (7.5x than gastroc., 9.5x than vasti, 24x than rect. fem., 9.5x than tib. ant., 12x than glut. max.; visual inspection of fig. 5). For the fore-aft acceleration a similar superior contribution of soleus was shown (3x higher than gastroc. and 8.8x higher than hamstrings [19]). The propulsive function of soleus during running is achieved by active shortening. Active shortening reduces the force-velocity potential as discussed extensively in the*

present manuscript. Consequently, a greater active muscle volume is required to achieve the required mechanical energy gain.

In contrast, the quadriceps muscle group, the main contributor during early stance, decelerating and supporting body mass [19,20], features more economical fascicle dynamics. Recently we showed that the fascicles of the vastus lateralis muscle, as a representative of the quadriceps muscle group operates with a high force-length (i.e. 0.91) and force-velocity potential (i.e. 0.97) during the stance phase of running. Operating at high force potentials reduces the energetic cost of this energetically expensive (due to its long fascicle length) muscle by reducing active muscle volume. This may indicate that the mechanical energy by muscular work required for steady state running is generated by muscles that are metabolically less expensive, likely to compensate for the reduction of the force-velocity potential.

We discussed the reviewers comment as a limitation in the revised manuscript (page: 8, line: 360):
“Although the soleus likely contributes to a great portion of the overall energetic costs during running, other limb muscles that were not considered in the present study are involved. However, the main energy source (positive work) is the ankle joint (41%) [17] and the soleus is the greatest muscle among the main plantar flexors with respect to physiological cross-sectional area (soleus 63%, gastroc. med. 25%, gastroc. lat. 12%) and volume (53%, 31% and 16% [18]). The key role of soleus is further supported by the modeling study of Hamner and Delp (2013), which showed that the soleus is by far the biggest contributor to the vertical acceleration and fore-aft acceleration of the center of mass [19]. This function is achieved by active shortening, which reduces the force-velocity potential and consequently requires a greater active muscle volume. In contrast, the quadriceps muscle group, the main contributor during early stance, decelerating and transferring body mass [19,20], features more economical fascicle dynamics. Recently we showed that the fascicles of the vastus lateralis muscle as a representative of the quadriceps muscle group operates with a high force-length (i.e. 0.91) and force-velocity potential (i.e. 0.97) during the stance phase of running. Operating at high force potentials minimizes the cost of this muscle, which is energetically expensive due to its long fascicle length (i.e. $L_0 = 94$ mm [15]), by reducing active muscle volume. This may indicate that the mechanical energy by muscular work required for steady state running is generated by muscles that are metabolically less expensive, likely to compensate for the reduction of the force-velocity potential.”

Comment: 2) The strict adherence to force-length and force-velocity relationships as defining features of muscle performance seems somewhat outdated given a wealth of literature showing that these relationships do not hold under conditions relevant to locomotion (i.e. history dependence, activation-dependent changes). These advances do not negate this study, however, it would be a more accurate representation of the field to acknowledge that they exist, and present this study as a means to investigate the importance of these relationships.

Response:

First note – History dependence:

We agree with the reviewers comment that the phenomenon of history dependence of force production after active muscle lengthening or shortening may be present for the soleus during running and may affect the force production [21–23]. In the present study, the soleus fascicles shortened continuously during running when activated, which would indicate a condition of force depression. Force depression has been shown to increase with increasing shortening magnitude [24], with decreasing shortening

velocity [25] and with increasing activation levels [26]. Since the soleus shortening magnitude was notable ($25.9 \pm 7.8 \%L_0$), the shortening velocity moderate ($0.118 \pm 0.039 V_{max}$) and the activation submaximal (average during stance phase: $0.32 \pm 0.19 EMG_{max}$; maximum activation: $0.52 \pm 0.18 EMG_{max}$), an effect of force depression on the force production can theoretically be expected. Yet, the force-length and force-velocity relationships remain the basic mechanisms for muscle force production. Interestingly, force depression is likely to be reduced due to the tendon and belly gearing mechanisms because those reduce the shortening magnitude and activation. The observed main finding of a correlation of the operating velocity and force-velocity potential with the energetic cost, however, does not neglect the presence of force depression but indicates that shortening velocity and consequently the force-velocity potential has a direct effect on the muscle energetics.

We added the following sentences in the introduction and discussion of the revised version of the manuscript (page: 2, line: 59; page: 7, line: 307):

“Besides the operating length and velocity as the main determinants, the history dependence of force generation [23], i.e. increased force after active muscle lengthening [27] and decreased force after active shortening [22,25], may additionally influence the force potential.”

“Furthermore, we showed that the soleus shortened continuously during the stance phase of running, which reflects a condition for force depression. Since a depression of force was shown to be accompanied by a decrease in the ATPase activity [28], force depression would have little or no effect on the energetic cost itself.”

Second note – Shift in optimal length:

Furthermore, it is correct that we assessed the force-length curve during maximal isometric contractions at different ankle joint angles and, using this relationship, we calculated the force-length potential of the soleus muscle during running at submaximal activation. There is evidence from early [29] and more recent [30,31] *in vitro* studies that the force-length curve depends on muscle activation, i.e. optimum length increases with submaximal activation. However, a recent study by Fontana and Herzog (2016) on the human vastus lateralis muscle showed that this holds not necessarily true for *in vivo* assessments [32]. In contrast to the *in vitro* studies, a rightward shift of optimal length was not observed when force was normalized to the maximum EMG signal (i.e. optimal length remained constant at different levels of activation). The authors suggested that the disagreement of the *in vitro* and *in vivo* studies might be an artefact related to the *in vitro* testing setup (e.g. non-physiological stimulation frequency range or calcium concentrations). Therefore, we can argue that mapping the submaximal fascicle operating length onto the force-length curve in the present *in vivo* study should not affect the findings.

We added the following information in the discussion part of revised manuscript as follows (page: 9, line: 385):

“Furthermore, we assessed the force-length curve during maximal isometric contractions and used it to calculate the force-length potential of the soleus muscle during running at submaximal activation. There is evidence from *in vitro* studies that the force-length curve depends on muscle activation [29–31]. However, in a recent *in vivo* study by Fontana and Herzog (2016) on the human vastus lateralis muscle, a rightward shift of optimal length with submaximal activation was not observed when force

was normalized to the maximum EMG signal [32]. The authors suggested that the shift in optimal length phenomenon might be related to the in vitro testing setup (e.g. non-physiological stimulation frequency range or Ca^{2+} concentrations). Therefore, we can argue that mapping the submaximal fascicle operating length onto the force-length curve in the present in vivo study should not affect the findings."

Comment: 3) This paper is fundamentally concerned with the effect of contractile conditions on muscle energy consumption. However, there is very little discussion of why length and velocity might affect energy consumption beyond required activation, despite a wealth of evidence on this i.e. how the cost per unit force varies across the force-length relationship in isolated muscle. In addition, it may be worth considering findings such as the effect of contractile history on cost (Joumaa et al., 2013), and the complexity of the cost of work (Holt et al., 2104; Curtin et al., 2019) in a more comprehensive discussion of in vivo muscle energetics.

Response:

First note - Variation of cost per unit force across the force-length relationship:

We agree with the reviewer that the energy turnover can differ across the force-length relationship in isolated animal muscle fibers tested in vitro [33]. During isometric contractions at sarcomere length shorter than optimal length, the force output is reduced but the ATPase rate seems not to greatly differ from the rate at optimal length, indicating a comparably higher cost of contraction at shorter length [34,35]. However, this effect seems to be more pronounced at very short lengths, which might not be covered during regular in vivo movements like locomotion, i.e. soleus operating range (0.75-1.01 L_0).

We added the following information in the revised version of the manuscript as follows (page: 7, line: 300): "Besides the favorable high force-length potential for economical force production, operating close to optimal length may additionally preserved from relatively higher energetic cost that can arise when contracting at shorter length. In vitro evidence showed that although force is reduced at shorter sarcomere length, the ATPase rate seems not to differ from the rate at optimal length, indicating comparably higher cost of contraction at shorter length [34,35]. However, this effect seems more pronounced at very short lengths, a portion of the force-length curve that is likely not covered by the soleus during running (operating range 0.75-1.01 L_0)."

Second note - Effect of contractile history on cost:

We added the following paragraph to the discussion part of the revised manuscript (page: 7, line: 307): "Furthermore, we showed that the soleus shortened continuously during the stance phase of running, which reflects a condition for force depression. Since a depression of force was shown to be accompanied by a decrease in the ATPase activity [28], force depression would have little or no effect on the energetic cost itself."

Third note - complexity of the cost of work:

Thanks for this comment. We agree with the reviewer on the ongoing debate on the cost of force and the cost of work. From our perspective, when a muscle contracts, force is generated and this consumes metabolic energy independently of the contraction type (i.e. isometric, eccentric, concentric). During

concentric contractions (active shortening) positive mechanical work is generated and during eccentric contractions (active lengthening) the work is negative. In stretch-shortening conditions, the net work could be zero when positive and negative work cancel each other out. Under isometric contractions, no mechanical work is generated by definition, which would again indicate no mechanical energy production (Joule), although force is generated and metabolic energy expended. The energy index of work in the context of the explanation of metabolic energy, therefore, might not be very appropriate (metabolic energy is not zero when work is zero e.g. during isometric contractions). Instead, an index of force and metabolic energy might better reflect the organismal cost during locomotion.

With our study, we cannot provide any new information on this discussion because work and force of soleus were not measured during running (which in our opinion is not possible at the moment). Therefore, we think that this topic is beyond of the scope of the present study and for this reason we would prefer not to go deeper in the discussion of cost of force and work but rather stay close to our experimental results.

Specific comments

Comment: Lines 50-51 – The ongoing debating between the cost of force and the cost of work as determinants of organismal cost should be acknowledged here. This could then also lead to a more nuanced discussion of factors dictating muscle energetics beyond simply level of activation.

Response: *As responded in more detail to the previous comment, we would not like to refer the manuscript to the discussion of cost of work and force because this is beyond the scope of the present study. By our study design (force and work not measured) and results we cannot provide any significant contribution to the mentioned ongoing discussion.*

Comment: Lines 123-124 and 185-187 – It is relatively unclear to me how the force-velocity relationship was determined here. It appears as though force and velocity were determined as fibers shortened against the tendon? Can the authors make this clearer, better define where in the contraction force and velocity were determined, and comment on how this might affect findings compared to a more standard isotonic or isovelocity protocol.

Response: *The force-velocity curve in the present study was not derived from experimentally measured force estimates and fascicle velocities. In the first version of the manuscript V_{max} was calculated based on the soleus muscle-specific constants of a_{rel} and b_{rel} reported by literature [10] as 11.75 L_0/s . According to a comment from the other reviewer, we now based our choice of V_{max} on more biological evidence as follows. The *in vitro* study of Luden et al., (2008) on the human soleus muscle reported V_{max} values for MHC I type fibers of 0.77 L_0/s and 2.91 L_0/s for MHC IIA type fibers measured at 15°C [3]. Considering the temperature coefficient provided by Ranatunga et al., (1984) [4] it can be predicted that V_{max} would increase to 4.4 L_0/s for MHC I type fibers and to 16.8 L_0/s for MHC IIA type fibers under physiological temperature conditions (37 °C). The fiber type distribution in the human soleus muscle can be estimated from literature reports, i.e. Johnson et al., 1973: type 1 fibers 87,7%, type 2 fibers (a and b) 12,3% (average of surface and deep fiber location) [5]; Larsson and Moss, 1993: type 1 89%, type 2A 11% [6]; Edgerton et al., 1975: slow twitch 70%, fast twitch 30% [7]; Luden et al.,*

2008: 74% MHC I, 20% MHC IIA (norm to 100%) [3]). Using an average of this reported distribution values (type 1: 81%, type 2: 19%), V_{max} for soleus under physiological temperature can be calculated as 6.77 L₀/s. The broad literature basis for the average fiber type distribution was also used to update a_{rel} to 0.175 (i.e. $0.1+0.4FT$, where FT is the fast twitch fiber type percentage [8,9]) and accordingly b_{rel} to 1.182 ($a_{rel} * V_{max}$ [10]). We then assessed the force-velocity curve by using the classical Hill formula, (i.e. $(F+a)(v+b)=(F_{max}+a)b$), and the muscle-specific values of V_{max} , a_{rel} and b_{rel} .

We recalculated the respective values (force-velocity potential and normalized velocities and their ranges) using the updated V_{max} of 6.77 L₀/s, a_{rel} and b_{rel} in the revised manuscript. Note that this adjustment in the calculation did not change any statistical result but only few numerical expressions (underlined in the revision). A revised and more detailed description of the calculation of the force-velocity potential is also now provided in the updated manuscript (see below).

The reason why we did not measure V_{max} experimentally is that precise measurements of V_{max} in vivo in humans are extremely challenging, technically and methodologically (e.g. restricted high dynamometer velocities, limited ultrasound capture frequencies in high velocities, limited range of motion to reach maximum force in high velocities, consideration of antagonistic co-contraction, mechanical properties of the tendon, history dependence effects).

We added the following information to the revised manuscript (page: 4, line: 155):

“Furthermore, we assessed the force-velocity relationship of soleus using the classical Hill equation [11] and the muscle-specific maximum fascicle shortening velocity (V_{max}) and constants of a_{rel} and b_{rel} . V_{max} was derived from the study of Luden et al. (2008), which showed V_{max} values for type 1 fibers of 0.77 L₀/s and 2.91 L₀/s for type 2 fibers of the human soleus muscle measured in vitro at 15°C [3]. Considering the temperature coefficient [4], V_{max} can be predicted as 4.4 L₀/s for type 1 fibers and 16.8 L₀/s for type 2 fibers under physiological temperature conditions (37 °C). Using an average fiber type distribution (type 1 fibers: 81%, type 2: 19%) of the human soleus muscle reported in literature [3,5–7], V_{max} can be calculated as 6.77 L₀/s. a_{rel} was calculated as $0.1+0.4FT$, where FT is the fast twitch fiber type percentage (see above), which then equals to 0.175 [8,9]. The product of a_{rel} and V_{max} then gives b_{rel} as 1.182 [10]. After rearrangement of the Hill formula and extension to the eccentric component, the normalized operating velocity (to V_{max}) can be used to calculate the individual force potential according to the force-velocity curve.”

Comment: Line 197-198 – The meaning of this is unclear to me. This description of touchdown and toe-off should be reworded for clarification.

Response: We changed the description to be more clear as follows (page: 4, line: 175):

“The touchdown of the foot and toe off were defined by the kinematic data as the first and second peak in knee extension, respectively [36,37].”

Comment: The results section is relatively dense. The authors may wish to consider moving some of the findings less critical to addressing their question to a table, to improve readability.

Response: Some of the results are now presented in the table to improve readability.

Comment: Line 299-300 – The assertion that the triceps surae consumes 40% of the cost during running is crucial to the argument of this paper. Yet it is not clear how this value is arrived at from the Fletcher and MacIntosh paper cited (the paper seems to give a large range of values for muscle energy consumption and not to relate this to organismal cost), and how reliable the output of their simple model is for this purpose. Could the authors give a little more detail on this (in the manuscript if of sufficient interest, or simply here). It may also be useful to combine this 40% estimate with the relative size of soleus to give a better representation of its likely contribution to energy consumption, considering fiber type as soleus is likely cheaper than gastrocs (Barclay, 1993).

Response: *The statement that the triceps surae consumes 40% of the energy during running can be derived from the comparison of figure 4 and 5 in the paper of Fletcher and MacIntosh (2015) and is numerically presented by the authors themselves in several subsequent published manuscripts (e.g. [38,39]). We agree with the reviewer that the presented calculations on muscle energy consumption in the aforementioned study may only provide a rough estimate. We also do not persist on the fixed value of 40% but rather we would like to understand this value as an indication of the great contribution of the triceps surae to the overall energetic cost. Within the triceps surae the gastrocnemius medialis and lateralis contribute to the propulsion as well but the physiological cross-sectional area (PCSA) and volume of soleus are notably higher (soleus: 131 cm² and 477 cm³, gastrocnemius medialis: 51 cm² and 285 cm³, gastrocnemius lateralis: 24 cm² and 146 cm³ [18]).*

Further calculations on the separate contribution of the single muscles of the triceps surae based on portions of force are very difficult if even possible because of strong underlying assumptions of the calculation. E.g., calculating the soleus muscle force using the PCSA relative to the other triceps muscles (gastroc. med and lat.) would premise that the force-potential due to the force length/velocity relationship and activation of all triceps surae muscles are equal. This assumption cannot be correct because the gastrocnemi are biarticular muscles. For this reason, we would not like to include this approach in our manuscript but rather stay on the more direct findings.

We softened our formulation by deleting the 40% in revised manuscript (page: 7, line: 280).

Comment: Lines 304-309 - The authors make a good case for why small changes in velocity would require an increase in activation and therefore cost. This effect should be seen in EMG recordings. It would seem that the argument could be strengthened by showing this as it would provide a more causal link between the change in muscle level function and organismal level cost.

Response: *Thanks for this comment. We did not go into any correlation analysis in the study because the parameter of surface EMG activation does not reflect active muscle volume adequately. However, a significant correlation can be found for the force-length-velocity potential (EMG mean: $r = -0.504$, $p = 0.028$; EMG max: $r = -0.525$, $p = 0.021$; EMG integral: $r = -0.504$, $p = 0.028$). Please note that the processing of the EMG signal can affect the correlation coefficients but not the significance itself ($p < 0.05$). Here a 20 Hz low pass filter was used after rectification and preprocessing with a high pass filter of 50 Hz.*

Given the mentioned limitation, the observed correlation might provide a cautious indication that a decreased EMG activity is associated with a higher force-length-velocity potential of the soleus muscle during the stance phase of running and that may affect the metabolic cost.

We added the association between EMG activity and force-length-velocity potential in the revised manuscript without an extended interpretation because, as we mentioned before, the active muscle volume cannot be assessed accurately from the EMG activity (page: 6, line: 255; page: 7, line: 291).

Comment: Line 334-335 – There seems to be some discrepancy regarding activation in here. The implication seems to be that muscle activation is higher in early stance to enable the tendon to be stretched, and then recoil to slow shortening velocity in the later part of stance. Yet a central claim of the paper is that cost is lower when shortening velocity is lower, due to a lower requirement for activation. It seems like the variation in required activation could balance out over the course of a stance phase? Could it be clarified as to why the early increase in activation to enable tendon stretch doesn't seem to be costly in the way that the latter reduction is deemed to be cheap?

Response: *Thanks for this comment. The rationale of this argumentation is that the observed activation pattern can be interpreted as appropriate for a coordinated MTU interaction during the running task with respect to economy. We changed the formulation in the respective section as follows (page: 7, line: 322):*

“The soleus produces mechanical work/energy for the lift and acceleration of the body throughout the entire stance phase. In the first half, where the MTU is elongated, the fascicles actively shorten. This means that a part of the mechanical energy of the human body is transferred to the tendon. Also, in this setting the muscle fascicles produce work under favorable conditions due to the force-length and force-velocity relationships (both potentials in this phase were very high) and save work as strain energy in the tendon. In the second half, the tendon strain energy is returned and at the same time the fascicles produce work by active shortening at a reduced force-velocity potential (fascicle shortening velocity is higher in this phase). The higher shortening velocity is associated with a reduction in the EMG activity and an increase in belly gearing. It has been suggested that increased gearing at fast shortening velocities and lower forces is a mechanism that allows particularly slower type fascicles to be more effective in generating forces [40]. This supports the idea that the observed activation pattern fostered an economical MTU interaction during running.”

Comment: Line 345 – typo “were”?

Response: *We corrected the typo accordingly.*

Comment: Line 375 – The study doesn't seem to show that energy consumption is related to the force-length-velocity potential, but rather just the force-velocity potential.

Response: *The force-length-velocity potential is the product of the force-length and force-velocity potential and was inversely associated with the energetic cost like the force-velocity potential. The force-length potential was consistently high among the participants and showed no significant association to the energetic cost. This indicates that the reason for the association of the force-length-velocity potential to the energetic cost was caused by the observed correlation of the force-velocity potential, i.e. variability in the force-length-velocity potential relied on the variability of the force-*

velocity potential that covered the variability of the energetic cost. However, as we mentioned in the discussion, a high force-length potential is also important for economical muscle force generation.

References used for the responses:

1. Epstein M. 1998 *Theoretical models of skeletal muscle: biological and mathematical considerations*. Chichester [u.a.]: Chichester ua : Wiley.
2. Zajac FE. 1989 Muscle and tendon: properties, models, scaling, and application to biomechanics and motor control. *Crit. Rev. Biomed. Eng.* **17**, 359–411.
3. Luden N, Minchev K, Hayes E, Louis E, Trappe T, Trappe S. 2008 Human vastus lateralis and soleus muscles display divergent cellular contractile properties. *Am. J. Physiol. - Regul. Integr. Comp. Physiol.* **295**, R1593–R1598. (doi:10.1152/ajpregu.90564.2008)
4. Ranatunga KW. 1984 The force-velocity relation of rat fast- and slow-twitch muscles examined at different temperatures. *J. Physiol.* **351**, 517–529.
5. Johnson MA, Polgar J, Weightman D, Appleton D. 1973 Data on the distribution of fibre types in thirty-six human muscles. An autopsy study. *J. Neurol. Sci.* **18**, 111–129. (doi:10.1016/0022-510x(73)90023-3)
6. Larsson L, Moss RL. 1993 Maximum velocity of shortening in relation to myosin isoform composition in single fibres from human skeletal muscles. *J. Physiol.* **472**, 595–614. (doi:10.1113/jphysiol.1993.sp019964)
7. Edgerton VR, Smith JL, Simpson DR. 1975 Muscle fibre type populations of human leg muscles. *Histochem. J.* **7**, 259–266.
8. Winters JM, Stark L. 1985 Analysis of Fundamental Human Movement Patterns Through the Use of In-Depth Antagonistic Muscle Models. *IEEE Trans. Biomed. Eng.* **BME-32**, 826–839. (doi:10.1109/TBME.1985.325498)
9. Winters JM, Stark L. 1988 Estimated mechanical properties of synergistic muscles involved in movements of a variety of human joints. *J. Biomech.* **21**, 1027–1041. (doi:10.1016/0021-9290(88)90249-7)
10. Miller RH, Umberger BR, Caldwell GE. 2012 Sensitivity of maximum sprinting speed to characteristic parameters of the muscle force–velocity relationship. *J. Biomech.* **45**, 1406–1413. (doi:10.1016/j.jbiomech.2012.02.024)
11. Hill Archibald Vivian. 1938 The heat of shortening and the dynamic constants of muscle. *Proc. R. Soc. Lond. Ser. B - Biol. Sci.* **126**, 136–195. (doi:10.1098/rspb.1938.0050)
12. Nikolaidou ME, Marzilger R, Bohm S, Mersmann F, Arampatzis A. 2017 Operating length and velocity of human M. vastus lateralis fascicles during vertical jumping. *R. Soc. Open Sci.* **4**, 170185. (doi:10.1098/rsos.170185)
13. Santuz A, Ekizos A, Janshen L, Baltzopoulos V, Arampatzis A. 2017 On the Methodological Implications of Extracting Muscle Synergies from Human Locomotion. *Int. J. Neural Syst.* **27**, 1750007. (doi:10.1142/S0129065717500071)
14. Marzilger R, Legerlotz K, Panteli C, Bohm S, Arampatzis A. 2018 Reliability of a semi-automated algorithm for the vastus lateralis muscle architecture measurement based on ultrasound images. *Eur. J. Appl. Physiol.* **118**, 291–301. (doi:10.1007/s00421-017-3769-8)
15. Bohm S, Marzilger R, Mersmann F, Santuz A, Arampatzis A. 2018 Operating length and velocity of human vastus lateralis muscle during walking and running. *Sci. Rep.* **8**, 5066. (doi:10.1038/s41598-018-23376-5)
16. Fletcher JR, MacIntosh BR. 2015 Achilles tendon strain energy in distance running: consider the muscle energy cost. *J. Appl. Physiol.* **118**, 193–199. (doi:10.1152/jappphysiol.00732.2014)
17. Novacheck TF. 1998 The biomechanics of running. *Gait Posture* **7**, 77–95. (doi:10.1016/S0966-6362(97)00038-6)
18. Albracht K, Arampatzis A, Baltzopoulos V. 2008 Assessment of muscle volume and physiological cross-sectional area of the human triceps surae muscle in vivo. *J. Biomech.* **41**, 2211–2218. (doi:10.1016/j.jbiomech.2008.04.020)
19. Hamner SR, Delp SL. 2013 Muscle contributions to fore-aft and vertical body mass center accelerations over a range of running speeds. *J. Biomech.* **46**, 780–787. (doi:10.1016/j.jbiomech.2012.11.024)
20. Dorn TW, Schache AG, Pandy MG. 2012 Muscular strategy shift in human running: dependence of running speed on hip and ankle muscle performance. *J. Exp. Biol.* **215**, 1944–1956. (doi:10.1242/jeb.064527)
21. Josephson RK. 1999 Dissecting muscle power output. *J. Exp. Biol.* **202**, 3369–3375.
22. Chen J, Hahn D, Power GA. 2019 Shortening-induced residual force depression in humans. *J. Appl. Physiol.* **126**, 1066–1073. (doi:10.1152/jappphysiol.00931.2018)

23. Herzog W. 2004 History dependence of skeletal muscle force production: implications for movement control. *Hum. Mov. Sci.* **23**, 591–604. (doi:10.1016/j.humov.2004.10.003)
24. Maréchal G, Plaghki L. 1979 The deficit of the isometric tetanic tension redeveloped after a release of frog muscle at a constant velocity. *J. Gen. Physiol.* **73**, 453–467.
25. Abbott BC, Aubert XM. 1952 The force exerted by active striated muscle during and after change of length. *J. Physiol.* **117**, 77–86. (doi:10.1113/jphysiol.1952.sp004733)
26. De Ruyter CJ, De Haan A, Jones DA, Sargeant AJ. 1998 Shortening-induced force depression in human adductor pollicis muscle. *J. Physiol.* **507**, 583–591. (doi:10.1111/j.1469-7793.1998.583bt.x)
27. Edman KA, Elzinga G, Noble MI. 1982 Residual force enhancement after stretch of contracting frog single muscle fibers. *J. Gen. Physiol.* **80**, 769–784. (doi:10.1085/jgp.80.5.769)
28. Joumaa V, Fitzowich A, Herzog W. 2017 Energy cost of isometric force production after active shortening in skinned muscle fibres. *J. Exp. Biol.* **220**, 1509–1515. (doi:10.1242/jeb.117622)
29. Rack PMH, Westbury DR. 1969 The effects of length and stimulus rate on tension in the isometric cat soleus muscle. *J. Physiol.* **204**, 443–460. (doi:10.1113/jphysiol.1969.sp008923)
30. Holt NC, Azizi E. 2014 What drives activation-dependent shifts in the force–length curve? *Biol. Lett.* **10**, 20140651. (doi:10.1098/rsbl.2014.0651)
31. Holt NC, Azizi E. 2016 The effect of activation level on muscle function during locomotion: are optimal lengths and velocities always used? *Proc R Soc B* **283**, 20152832. (doi:10.1098/rspb.2015.2832)
32. Fontana H de B, Herzog W. 2016 Vastus lateralis maximum force-generating potential occurs at optimal fascicle length regardless of activation level. *Eur. J. Appl. Physiol.* **116**, 1267–1277. (doi:10.1007/s00421-016-3381-3)
33. Barclay CJ. 2015 Energetics of contraction. *Compr. Physiol.* **5**, 961–995. (doi:10.1002/cphy.c140038)
34. Stephenson DG, Stewart AW, Wilson GJ. 1989 Dissociation of force from myofibrillar MgATPase and stiffness at short sarcomere lengths in rat and toad skeletal muscle. *J. Physiol.* **410**, 351–366. (doi:10.1113/jphysiol.1989.sp017537)
35. Hilber K, Sun Y-B, Irving M. 2001 Effects of sarcomere length and temperature on the rate of ATP utilisation by rabbit psoas muscle fibres. *J. Physiol.* **531**, 771–780. (doi:10.1111/j.1469-7793.2001.0771h.x)
36. Smith L, Preece S, Mason D, Bramah C. 2015 A comparison of kinematic algorithms to estimate gait events during overground running. *Gait Posture* **41**, 39–43. (doi:10.1016/j.gaitpost.2014.08.009)
37. Fellin RE, Rose WC, Royer TD, Davis IS. 2010 Comparison of methods for kinematic identification of footstrike and toe-off during overground and treadmill running. *J. Sci. Med. Sport* **13**, 646–650. (doi:10.1016/j.jsams.2010.03.006)
38. Fletcher JR, MacIntosh BR. 2018 Theoretical considerations for muscle-energy savings during distance running. *J. Biomech.* **73**, 73–79. (doi:10.1016/j.jbiomech.2018.03.023)
39. Fletcher JR, MacIntosh BR. 2017 Running Economy from a Muscle Energetics Perspective. *Front. Physiol.* **8**, 433. (doi:10.3389/fphys.2017.00433)
40. Wakeling JM, Blake OM, Wong I, Rana M, Lee SSM. 2011 Movement mechanics as a determinate of muscle structure, recruitment and coordination. *Philos. Trans. R. Soc. Lond. B Biol. Sci.* **366**, 1554–1564. (doi:10.1098/rstb.2010.0294)

Appendix C

Referee: 2

I appreciate the authors addressing these comments and amending the manuscript accordingly.

A few final responses to the authors' responses are made in red below

Comment: In this study, the authors explore the effect of muscle force-length and force-velocity conditions on the energetic cost of running. In addition, they explore the determinants of muscle fiber length change. I find this to be an extensive and well collected data set that uses in vivo determination of force-length and force-velocity relationships, and application of these to muscle function during running to address these questions. There appear to me to be a few major limitations of the study. These should be addressed throughout.

Response: *Thank you for your valuable comments. All changes are underlined in the revised version of the manuscript and the references cited in the responses can be found at the end of the document. Please note that some parts of the methods are now presented in the electronic supplementary material due to length restrictions of the journal.*

Comment: 1) Organismal energy consumption is measured, but only the length and velocity profile of soleus. This prevents the authors from drawing the more interesting conclusion that muscle shortening velocity is a determinant of energy consumption. This is acknowledged and somewhat addressed in the discussion, but remains a major limitation to the study.

Response: *We agree with the reviewers comment that although the soleus contributes to a great portion of the overall energetic costs during running [16] other limb muscles are involved that were not considered in the present study and this remains a limitation. However, the main energy source (positive work) is the ankle joint (41%) while the contribution of the knee and hip joint is comparably lower during running [17]. The soleus is the greatest muscle among the main plantar flexors with respect to physiological cross-sectional area and volume (soleus: 131 cm² and 477 cm³, gastrocnemius medialis: 51 cm² and 285 cm³, gastrocnemius lateralis: 24 cm² and 146 cm³ [18]), giving this muscle a key role. Using a modeling approach, Hamner and Delp (2013) showed that the contribution of soleus to the vertical acceleration of the center of mass at the same velocity as in our study (i.e. 2.5 m/s) is remarkably higher than those of the other lower limb muscles (7.5x than gastroc., 9.5x than vasti, 24x than rect. fem., 9.5x than tib. ant., 12x than glut. max.; visual inspection of fig. 5). For the fore-aft acceleration a similar superior contribution of soleus was shown (3x higher than gastroc. and 8.8x higher than hamstrings [19]). The propulsive function of soleus during running is achieved by active shortening. Active shortening reduces the force-velocity potential as discussed extensively in the present manuscript. Consequently, a greater active muscle volume is required to achieve the required mechanical energy gain.*

In contrast, the quadriceps muscle group, the main contributor during early stance, decelerating and supporting body mass [19,20], features more economical fascicle dynamics. Recently we showed that the fascicles of the vastus lateralis muscle, as a representative of the quadriceps muscle group operates with a high force-length (i.e. 0.91) and force-velocity potential (i.e. 0.97) during the stance phase of running. Operating at high force potentials reduces the energetic cost of this energetically expensive (due to its long fascicle length) muscle by reducing active muscle volume. This may indicate that the

mechanical energy by muscular work required for steady state running is generated by muscles that are metabolically less expensive, likely to compensate for the reduction of the force-velocity potential.

We discussed the reviewers comment as a limitation in the revised manuscript (page: 8, line: 360):
“Although the soleus likely contributes to a great portion of the overall energetic costs during running, other limb muscles that were not considered in the present study are involved. However, the main energy source (positive work) is the ankle joint (41%) [17] and the soleus is the greatest muscle among the main plantar flexors with respect to physiological cross-sectional area (soleus 63%, gastroc. med. 25%, gastroc. lat. 12%) and volume (53%, 31% and 16% [18]). The key role of soleus is further supported by the modeling study of Hamner and Delp (2013), which showed that the soleus is by far the biggest contributor to the vertical acceleration and fore-aft acceleration of the center of mass [19]. This function is achieved by active shortening, which reduces the force-velocity potential and consequently requires a greater active muscle volume. In contrast, the quadriceps muscle group, the main contributor during early stance, decelerating and transferring body mass [19,20], features more economical fascicle dynamics. Recently we showed that the fascicles of the vastus lateralis muscle as a representative of the quadriceps muscle group operates with a high force-length (i.e. 0.91) and force-velocity potential (i.e. 0.97) during the stance phase of running. Operating at high force potentials minimizes the cost of this muscle, which is energetically expensive due to its long fascicle length (i.e. $L_0 = 94$ mm [15]), by reducing active muscle volume. This may indicate that the mechanical energy by muscular work required for steady state running is generated by muscles that are metabolically less expensive, likely to compensate for the reduction of the force-velocity potential.”

Comment: 2) The strict adherence to force-length and force-velocity relationships as defining features of muscle performance seems somewhat outdated given a wealth of literature showing that these relationships do not hold under conditions relevant to locomotion (i.e. history dependence, activation-dependent changes). These advances do not negate this study, however, it would be a more accurate representation of the field to acknowledge that they exist, and present this study as a means to investigate the importance of these relationships.

Response:

First note – History dependence:

We agree with the reviewers comment that the phenomenon of history dependence of force production after active muscle lengthening or shortening may be present for the soleus during running and may affect the force production [21–23]. In the present study, the soleus fascicles shortened continuously during running when activated, which would indicate a condition of force depression. Force depression has been shown to increase with increasing shortening magnitude [24], with decreasing shortening velocity [25] and with increasing activation levels [26]. Since the soleus shortening magnitude was notable (25.9 ± 7.8 % L_0), the shortening velocity moderate (0.118 ± 0.039 V_{max}) and the activation submaximal (average during stance phase: 0.32 ± 0.19 EMG_{max} ; maximum activation: 0.52 ± 0.18 EMG_{max}), an effect of force depression on the force production can theoretically be expected. Yet, the force-length and force-velocity relationships remain the basic mechanisms for muscle force production. Interestingly, force depression is likely to be reduced due to the tendon and belly gearing mechanisms because those reduce the shortening magnitude and activation. The observed main finding of a

correlation of the operating velocity and force-velocity potential with the energetic cost, however, does not neglect the presence of force depression but indicates that shortening velocity and consequently the force-velocity potential has a direct effect on the muscle energetics.

We added the following sentences in the introduction and discussion of the revised version of the manuscript (page: 2, line: 59; page: 7, line: 307):

“Besides the operating length and velocity as the main determinants, the history dependence of force generation [23], i.e. increased force after active muscle lengthening [27] and decreased force after active shortening [22,25], may additionally influence the force potential.”

“Furthermore, we showed that the soleus shortened continuously during the stance phase of running, which reflects a condition for force depression. Since a depression of force was shown to be accompanied by a decrease in the ATPase activity [28], force depression would have little or no effect on the energetic cost itself.”

Second note – Shift in optimal length:

Furthermore, it is correct that we assessed the force-length curve during maximal isometric contractions at different ankle joint angles and, using this relationship, we calculated the force-length potential of the soleus muscle during running at submaximal activation. There is evidence from early [29] and more recent [30,31] in vitro studies that the force-length curve depends on muscle activation, i.e. optimum length increases with submaximal activation. However, a recent study by Fontana and Herzog (2016) on the human vastus lateralis muscle showed that this holds not necessarily true for in vivo assessments [32]. In contrast to the in vitro studies, a rightward shift of optimal length was not observed when force was normalized to the maximum EMG signal (i.e. optimal length remained constant at different levels of activation). The authors suggested that the disagreement of the in vitro and in vivo studies might be an artefact related to the in vitro testing setup (e.g. non-physiological stimulation frequency range or calcium concentrations). Therefore, we can argue that mapping the submaximal fascicle operating length onto the force-length curve in the present in vivo study should not affect the findings.

We added the following information in the discussion part of revised manuscript as follows (page: 9, line: 385):

“Furthermore, we assessed the force-length curve during maximal isometric contractions and used it to calculate the force-length potential of the soleus muscle during running at submaximal activation. There is evidence from in vitro studies that the force-length curve depends on muscle activation [29–31]. However, in a recent in vivo study by Fontana and Herzog (2016) on the human vastus lateralis muscle, a rightward shift of optimal length with submaximal activation was not observed when force was normalized to the maximum EMG signal [32]. The authors suggested that the shift in optimal length phenomenon might be related to the in vitro testing setup (e.g. non-physiological stimulation frequency range or Ca²⁺ concentrations). Therefore, we can argue that mapping the submaximal fascicle operating length onto the force-length curve in the present in vivo study should not affect the findings.”

I find the rationale of this Herzog and Fontana paper quite difficult to follow, particular with regards to how they cite other studies. They attempt to distinguish between activation and force which, while

may have some bearing on the mechanism responsible, does nothing to counter the finding that shifts in optimum length not predicted by the sliding filament theory occur with changing contractile conditions. For the purposes of this argument, what I take from this study is that if you change muscle force production, optimum length shifts (Fig. 3 Herzog and Fontana 2016). This is in line with other *in vivo* studies of this phenomenon (Ichinose et al., 1997; Kwah et al 2013), and makes it less obvious that this potential effect should be ignored in the present paper.

This does not change the arguments of this paper, and I leave this to the authors discretion, but it is my feeling that it would be a stronger paper if it shifted its focus from the conviction that force-length and force-velocity potential of the muscle dictates *in vivo* performance (if this were true, Hill-type muscle models would do a better job (Lee et al., 2013; Dick et al., 2017)) to an argument that multiple factors influence muscle mechanical and energetic performance under dynamic conditions, and that this paper seeks to understand to what extent force-velocity effects dictate energetic performance. It's a subtle shift that would require minor rewording throughout, but it's my feeling that this would much better reflect that state of the field.

Reference to the Herzog and Fontana paper is made again in the discussion, the authors may wish to consider how well it supports their argument and the contradictory findings of other *in vivo* papers.

Comment: 3) This paper is fundamentally concerned with the effect of contractile conditions on muscle energy consumption. However, there is very little discussion of why length and velocity might affect energy consumption beyond required activation, despite a wealth of evidence on this i.e. how the cost per unit force varies across the force-length relationship in isolated muscle. In addition, it may be worth considering findings such as the effect of contractile history on cost (Joumaa et al., 2013), and the complexity of the cost of work (Holt et al., 2104; Curtin et al., 2019) in a more comprehensive discussion of *in vivo* muscle energetics.

Response:

First note - Variation of cost per unit force across the force-length relationship:

*We agree with the reviewer that the energy turnover can differ across the force-length relationship in isolated animal muscle fibers tested *in vitro* [33]. During isometric contractions at sarcomere length shorter than optimal length, the force output is reduced but the ATPase rate seems not to greatly differ from the rate at optimal length, indicating a comparably higher cost of contraction at shorter length [34,35]. However, this effect seems to be more pronounced at very short lengths, which might not be covered during regular *in vivo* movements like locomotion, i.e. soleus operating range (0.75-1.01 L_0).*

*We added the following information in the revised version of the manuscript as follows (page: 7, line: 300): "Besides the favorable high force-length potential for economical force production, operating close to optimal length may additionally preserved from relatively higher energetic cost that can arise when contracting at shorter length. *In vitro* evidence showed that although force is reduced at shorter sarcomere length, the ATPase rate seems not to differ from the rate at optimal length, indicating comparably higher cost of contraction at shorter length [34,35]. However, this effect seems more pronounced at very short lengths, a portion of the force-length curve that is likely not covered by the soleus during running (operating range 0.75-1.01 L_0)."*

Second note - Effect of contractile history on cost:

We added the following paragraph to the discussion part of the revised manuscript (page: 7, line: 307):
“Furthermore, we showed that the soleus shortened continuously during the stance phase of running, which reflects a condition for force depression. Since a depression of force was shown to be accompanied by a decrease in the ATPase activity [28], force depression would have little or no effect on the energetic cost itself.”

Third note - complexity of the cost of work:

Thanks for this comment. We agree with the reviewer on the ongoing debate on the cost of force and the cost of work. From our perspective, when a muscle contracts, force is generated and this consumes metabolic energy independently of the contraction type (i.e. isometric, eccentric, concentric). During concentric contractions (active shortening) positive mechanical work is generated and during eccentric contractions (active lengthening) the work is negative. In stretch-shortening conditions, the net work could be zero when positive and negative work cancel each other out. Under isometric contractions, no mechanical work is generated by definition, which would again indicate no mechanical energy production (Joule), although force is generated and metabolic energy expended. The energy index of work in the context of the explanation of metabolic energy, therefore, might not be very appropriate (metabolic energy is not zero when work is zero e.g. during isometric contractions). Instead, an index of force and metabolic energy might better reflect the organismal cost during locomotion.

With our study, we cannot provide any new information on this discussion because work and force of soleus were not measured during running (which in our opinion is not possible at the moment). Therefore, we think that this topic is beyond of the scope of the present study and for this reason we would prefer not to go deeper in the discussion of cost of force and work but rather stay close to our experimental results.

This cost of work argument could entirely be thought of as cost of muscle fiber shortening argument. Which is obviously very pertinent to this paper. It is my opinion that this paper would be strengthened by greater discussion of this complexity and what the data presented here do to advance our understanding -i.e. cheap work (shortening) may be possible in some cases (Holt et al., 2014; Curtin et al., 2019), but in this case, more rapid active muscle shortening does seem to incur energetic costs. But again, I leave this to the authors discretion.

Specific comments

Comment: Lines 50-51 – The ongoing debating between the cost of force and the cost of work as determinants of organismal cost should be acknowledged here. This could then also lead to a more nuanced discussion of factors dictating muscle energetics beyond simply level of activation.

Response: As responded in more detail to the previous comment, we would not like to refer the manuscript to the discussion of cost of work and force because this is beyond the scope of the present study. By our study design (force and work not measured) and results we cannot provide any significant contribution to the mentioned ongoing discussion.

Comment: Lines 123-124 and 185-187 – It is relatively unclear to me how the force-velocity relationship was determined here. It appears as though force and velocity were determined as fibers shortened against the tendon? Can the authors make this clearer, better define where in the contraction force and velocity were determined, and comment on how this might affect findings compared to a more standard isotonic or isovelocity protocol.

Response: *The force-velocity curve in the present study was not derived from experimentally measured force estimates and fascicle velocities. In the first version of the manuscript V_{max} was calculated based on the soleus muscle-specific constants of a_{rel} and b_{rel} reported by literature [10] as 11.75 L_0/s . According to a comment from the other reviewer, we now based our choice of V_{max} on more biological evidence as follows. The in vitro study of Luden et al., (2008) on the human soleus muscle reported V_{max} values for MHC I type fibers of 0.77 L_0/s and 2.91 L_0/s for MHC IIA type fibers measured at 15°C [3]. Considering the temperature coefficient provided by Ranatunga et al., (1984) [4] it can be predicted that V_{max} would increase to 4.4 L_0/s for MHC I type fibers and to 16.8 L_0/s for MHC IIA type fibers under physiological temperature conditions (37 °C). The fiber type distribution in the human soleus muscle can be estimated from literature reports, i.e. Johnson et al., 1973: type 1 fibers 87,7%, type 2 fibers (a and b) 12,3% (average of surface and deep fiber location) [5]; Larsson and Moss, 1993: type 1 89%, type 2A 11% [6]; Edgerton et al., 1975: slow twitch 70%, fast twitch 30% [7]; Luden et al., 2008: 74% MHC I, 20% MHC IIA (norm to 100%) [3]. Using an average of this reported distribution values (type 1: 81%, type 2: 19%), V_{max} for soleus under physiological temperature can be calculated as 6.77 L_0/s . The broad literature basis for the average fiber type distribution was also used to update a_{rel} to 0.175 (i.e. $0.1+0.4FT$, where FT is the fast twitch fiber type percentage [8,9]) and accordingly b_{rel} to 1.182 ($a_{rel} * V_{max}$ [10]). We then assessed the force-velocity curve by using the classical Hill formula, (i.e. $(F+a)(v+b)=(F_{max}+a)b$), and the muscle-specific values of V_{max} , a_{rel} and b_{rel} .*

We recalculated the respective values (force-velocity potential and normalized velocities and their ranges) using the updated V_{max} of 6.77 L_0/s , a_{rel} and b_{rel} in the revised manuscript. Note that this adjustment in the calculation did not changed any statistical result but only few numerical expressions (underlined in the revision). A revised and more detailed description of the calculation of the force-velocity potential is also now provided in the updated manuscript (see below).

The reason why we did not measured V_{max} experimentally is that precise measurements of V_{max} in vivo in humans are extremely challenging, technically and methodologically (e.g. restricted high dynamometer velocities, limited ultrasound capture frequencies in high velocities, limited range of motion to reach maximum force in high velocities, consideration of antagonistic co-contraction, mechanical properties of the tendon, history dependence effects).

We added the following information to the revised manuscript (page: 4, line: 155):

“Furthermore, we assessed the force-velocity relationship of soleus using the classical Hill equation [11] and the muscle-specific maximum fascicle shortening velocity (V_{max}) and constants of a_{rel} and b_{rel} . V_{max} was derived from the study of Luden et al. (2008), which showed V_{max} values for type 1 fibers of 0.77 L_0/s and 2.91 L_0/s for type 2 fibers of the human soleus muscle measured in vitro at 15°C [3]. Considering the temperature coefficient [4], V_{max} can be predicted as 4.4 L_0/s for type 1 fibers and 16.8 L_0/s for type 2 fibers under physiological temperature conditions (37 °C). Using an average fiber type distribution (type 1 fibers: 81%, type 2: 19%) of the human soleus muscle reported in literature [3,5–7], V_{max} can be calculated as 6.77 L_0/s . a_{rel} was calculated as $0.1+0.4FT$, where FT is the fast twitch fiber type percentage (see above), which then equals to 0.175 [8,9]. The product of a_{rel} and V_{max} then gives b_{rel} as 1.182 [10]. After rearrangement of the Hill formula and extension to the eccentric component,

the normalized operating velocity (to V_{max}) can be used to calculate the individual force potential according to the force-velocity curve."

Line 104-103 – it therefore seems misleading to say 'as a function of their experimentally assessed force-velocity relationships'

Line 126-127 – similar issue in that this seems to suggest experimental measurements of force-velocity relationships in this study

Comment: Line 197-198 – The meaning of this is unclear to me. This description of touchdown and toe-off should be reworded for clarification.

Response: *We changed the description to be more clear as follows (page: 4, line: 175):*
"The touchdown of the foot and toe off were defined by the kinematic data as the first and second peak in knee extension, respectively [36,37]."

Comment: The results section is relatively dense. The authors may wish to consider moving some of the findings less critical to addressing their question to a table, to improve readability.

Response: *Some of the results are now presented in the table to improve readability.*

Comment: Line 299-300 – The assertion that the triceps surae consumes 40% of the cost during running is crucial to the argument of this paper. Yet it is not clear how this value is arrived at from the Fletcher and MacIntosh paper cited (the paper seems to give a large range of values for muscle energy consumption and not to relate this to organismal cost), and how reliable the output of their simple model is for this purpose. Could the authors give a little more detail on this (in the manuscript if of sufficient interest, or simply here). It may also be useful to combine this 40% estimate with the relative size of soleus to give a better representation of its likely contribution to energy consumption, considering fiber type as soleus is likely cheaper than gastrocs (Barclay, 1993).

Response: *The statement that the triceps surae consumes 40% of the energy during running can be derived from the comparison of figure 4 and 5 in the paper of Fletcher and MacIntosh (2015) and is numerically presented by the authors themselves in several subsequent published manuscripts (e.g. [38,39]). We agree with the reviewer that the presented calculations on muscle energy consumption in the aforementioned study may only provide a rough estimate. We also do not persist on the fixed value of 40% but rather we would like to understand this value as an indication of the great contribution of the triceps surae to the overall energetic cost. Within the triceps surae the gastrocnemius medialis and lateralis contribute to the propulsion as well but the physiological cross-sectional area (PCSA) and volume of soleus are notably higher (soleus: 131 cm² and 477 cm³, gastrocnemius medialis: 51 cm² and 285 cm³, gastrocnemius lateralis: 24 cm² and 146 cm³ [18]).*

Further calculations on the separate contribution of the single muscles of the triceps surae based on portions of force are very difficult if even possible because of strong underlying assumptions of the calculation. E.g., calculating the soleus muscle force using the PCSA relative to the other triceps muscles (gastroc. med and lat.) would premise that the force-potential due to the force length/velocity relationship and activation of all triceps surae muscles are equal. This assumption cannot be correct

because the gastrocnemii are biarticular muscles. For this reason, we would not like to include this approach in our manuscript but rather stay on the more direct findings.

We softened our formulation by deleting the 40% in revised manuscript (page: 7, line: 280).

Comment: Lines 304-309 - The authors make a good case for why small changes in velocity would require an increase in activation and therefore cost. This effect should be seen in EMG recordings. It would seem that the argument could be strengthened by showing this as it would provide a more causal link between the change in muscle level function and organismal level cost.

Response: *Thanks for this comment. We did not go into any correlation analysis in the study because the parameter of surface EMG activation does not reflect active muscle volume adequately. However, a significant correlation can be found for the force-length-velocity potential (EMG mean: $r = -0.504$, $p = 0.028$; EMG max: $r = -0.525$, $p = 0.021$; EMG integral: $r = -0.504$, $p = 0.028$). Please note that the processing of the EMG signal can affect the correlation coefficients but not the significance itself ($p < 0.05$). Here a 20 Hz low pass filter was used after rectification and preprocessing with a high pass filter of 50 Hz.*

Given the mentioned limitation, the observed correlation might provide a cautious indication that a decreased EMG activity is associated with a higher force-length-velocity potential of the soleus muscle during the stance phase of running and that may affect the metabolic cost.

We added the association between EMG activity and force-length-velocity potential in the revised manuscript without an extended interpretation because, as we mentioned before, the active muscle volume cannot be assessed accurately from the EMG activity (page: 6, line: 255; page: 7, line: 291).

Comment: Line 334-335 – There seems to be some discrepancy regarding activation in here. The implication seems to be that muscle activation is higher in early stance to enable the tendon to be stretched, and then recoil to slow shortening velocity in the later part of stance. Yet a central claim of the paper is that cost is lower when shortening velocity is lower, due to a lower requirement for activation. It seems like the variation in required activation could balance out over the course of a stance phase? Could it be clarified as to why the early increase in activation to enable tendon stretch doesn't seem to be costly in the way that the latter reduction is deemed to be cheap?

Response: *Thanks for this comment. The rationale of this argumentation is that the observed activation pattern can be interpreted as appropriate for a coordinated MTU interaction during the running task with respect to economy. We changed the formulation in the respective section as follows (page: 7, line: 322):*

“The soleus produces mechanical work/energy for the lift and acceleration of the body throughout the entire stance phase. In the first half, where the MTU is elongated, the fascicles actively shorten. This means that a part of the mechanical energy of the human body is transferred to the tendon. Also, in this setting the muscle fascicles produce work under favorable conditions due to the force-length and force-velocity relationships (both potentials in this phase were very high) and save work as strain energy in the tendon. In the second half, the tendon strain energy is returned and at the same time the fascicles produce work by active shortening at a reduced force-velocity potential (fascicle shortening

velocity is higher in this phase). The higher shortening velocity is associated with a reduction in the EMG activity and an increase in belly gearing. It has been suggested that increased gearing at fast shortening velocities and lower forces is a mechanism that allows particularly slower type fascicles to be more effective in generating forces [40]. This supports the idea that the observed activation pattern fostered an economical MTU interaction during running.”

Comment: Line 345 – typo “were”?

Response: We corrected the typo accordingly.

Comment: Line 375 – The study doesn’t seem to show that energy consumption is related to the force-length-velocity potential, but rather just the force-velocity potential.

Response: *The force-length-velocity potential is the product of the force-length and force-velocity potential and was inversely associated with the energetic cost like the force-velocity potential. The force-length potential was consistently high among the participants and showed no significant association to the energetic cost. This indicates that the reason for the association of the force-length-velocity potential to the energetic cost was caused by the observed correlation of the force-velocity potential, i.e. variability in the force-length-velocity potential relied on the variability of the force-velocity potential that cohered the variability of the energetic cost. However, as we mentioned in the discussion, a high force-length potential is also important for economical muscle force generation.*

Appendix D

Response to referees

Response to referee 1

Comment: The authors have addressed all my previous concerns in a careful manner.

Response: *Once again thank you for your valuable review.*

Comment: There remains one further comment that they may choose to consider for the manuscript, and it still concerns the choice of V_{max} . I appreciate the further analysis that the authors have attempted, to provide a value of V_{max} for the Soleus. However, it should be noted that the running velocity of 2.5 m/s is not all that fast, and indeed the EMG averages less than 50%. As such, it is likely that the fastest muscle fibres will not have been recruited, and hence the weighted mean taken for V_{max} may thus be an overestimate. Coupled to this, with more than half of the muscle inactive, the actual V_{max} may be less than its constituent fibres (for additional reasons: Holt et al. Proc Roy Soc B 2014). If the V_{max} for the Soleus were less than the estimated 6.77 L/s for this experimental situation, then it is likely that the actual spread of Force-velocity potentials would be larger than shown in Fig. 3. It is thus worth considering that you have actually resulted with a conservative evaluation of the importance of the force-velocity potential.

Response: *Thank you for this comment. We agree with the opinion of the reviewer that V_{max} during submaximal running in vivo might be influenced by factors not considered in our calculation (e.g. selective slow fiber type recruitment, muscle resistance to shortening). Referring to the results of Holt et al. (2014), V_{max} could be in deed lower. We now acknowledge this aspect in the respective paragraph of the revised manuscript (see below, page: 8, line: 378). We also added the aspect of more economical selective slow fiber type recruitment for submaximal slow contractions as during running to a more comprehensive paragraph in the discussion section of the revised manuscript (see response to reviewer 2, page: 9, line: 399).*

“To assess the force-velocity potential we used a biologically funded value of V_{max} , based on in vitro studies human soleus, i.e. $6.77 L_0 s^{-1}$ ($279.0 \pm 34.9 mm s^{-1}$). However, during submaximal running in vivo the lower activation level and selective slow fiber type recruitment may affect the actual force-velocity potential of the soleus muscle.”

Response to referee 2

General: *Once again thank you for your valuable review.*

Comment: I find the rationale of this Herzog and Fontana paper quite difficult to follow, particular with regards to how they cite other studies. They attempt to distinguish between activation and force which, while may have some bearing on the mechanism responsible, does nothing to counter the

finding that shifts in optimum length not predicted by the sliding filament theory occur with changing contractile conditions. For the purposes of this argument, what I take from this study is that if you change muscle force production, optimum length shifts (Fig. 3 Herzog and Fontana 2016). This is in line with other in vivo studies of this phenomenon (Ichinose et al., 1997; Kwah et al 2013), and makes it less obvious that this potential effect should be ignored in the present paper.

This does not change the arguments of this paper, and I leave this to the authors discretion, but it is my feeling that it would be a stronger paper if it shifted its focus from the conviction that force-length and force-velocity potential of the muscle dictates in vivo performance (if this were true, Hill-type muscle models would do a better job (Lee et al., 2013; Dick et al., 2017)) to an argument that multiple factors influence muscle mechanical and energetic performance under dynamic conditions, and that this paper seeks to understand to what extent force-velocity effects dictate energetic performance. It's a subtle shift that would require minor rewording throughout, but it's my feeling that this would much better reflect that state of the field. Reference to the Herzog and Fontana paper is made again in the discussion, the authors may wish to consider how well it supports their argument and the contradictory findings of other in vivo papers.

Response:

First note: Fontana and Herzog (2016) paper and lack of shift in optimal length

As described by Fontana and Herzog (2016) – and we agree on that – the shift in optimal length reported in the former human in vivo study of Ichinose et al., (1997) is constrained by the experimental setup because the authors controlled the torque (i.e. used a percentage of the maximum force) and not the muscle activation in each of the assessed knee joint angles. Due to the force-dependent elongation of tendon and aponeurosis, the result of a shift in optimal fascicle length is to be expected and the conclusion of a shift in optimal length is misleading. The experimental constrain from the Ichinose et al., (1997) study was overcome in the Fontana and Herzog (2016) study by referring the fascicle length to activation level, leading to the lack of shift in optimal length.

We added the following text in the revised manuscript (page: 9, line: 395):

“The discrepancy of the in vitro and in vivo evidence clearly warrants future investigation to elucidate the shifting length phenomenon in the context of in vivo submaximal locomotion. Given the current human in vivo evidence [1], we can argue that mapping the submaximal fascicle operating length onto the force-length curve in the present in vivo study should not affect the findings.”

Second note: Complexity of energetic cost and muscle contraction

We agree with the opinion of the reviewer that multiple factors may affect energetic cost during submaximal human running and further that simple models not reflect the complexity of muscle mechanical and energetic performance under dynamic conditions appropriately. To address the reviewers general comment we added the following paragraph to the limitations section of the revised manuscript (page: 9, line: 400):

“In the present study we focused on the understanding of the contribution of the force-length and force-velocity potential to the energetic cost of running and we showed that the force-velocity potential is inversely related to the energetic cost, explaining about one third of its variance. We argue that an increase of active muscle volume due to the decreased force-velocity potential would increase the energetic cost of running. However, it must be acknowledged that the energetic cost of muscle

contraction is complex and multifactorial. Independent of active muscle volume, in higher shortening velocities the rate of cross-bridges cycling is increased and as a consequence the consumed energy. In our study, shortening velocities of the soleus muscle were in average $0.118 V_{\max}$ throughout the stance phase, a range where the rate of ATP hydrolysis shows a steep increase [2]. Furthermore, in submaximal intensity contractions as during our investigated running velocity selective slow fiber type activation might decrease the energetic cost by reducing the contribution of energetically more expensive fast twitch fibers.”

Comment: This cost of work argument could entirely be thought of as cost of muscle fiber shortening argument. Which is obviously very pertinent to this paper. It is my opinion that this paper would be strengthened by greater discussion of this complexity and what the data presented here do to advance our understanding -i.e. cheap work (shortening) may be possible in some cases (Holt et al., 2014; Curtin et al., 2019), but in this case, more rapid active muscle shortening does seem to incur energetic costs. But again, I leave this to the authors discretion.

Response: *Thanks for this comment. We added the aforementioned paragraph to the manuscript to provide a broader discussion of this topic.*

Comment: Line 104-103 – it therefore seems misleading to say ‘as a function of their experimentally assessed force-velocity relationships’. Line 126-127 – similar issue in that this seems to suggest experimental measurements of force-velocity relationships in this study.

Response: *To avoid any confusion we reworded the sentences as follows:*

“In the present study, we investigated the operating length and velocity of the soleus muscle fascicles (i.e. bundles of fibers) during running as a function of the experimentally determined force-length and assessed force-velocity relationships (i.e. force-length and force-velocity potential) and their association to the energetic cost of running. “

“The derived optimal fascicle length for force production was further used to calculate the force-velocity relationship of the soleus fascicles.”

References

1. Fontana H de B, Herzog W. 2016 Vastus lateralis maximum force-generating potential occurs at optimal fascicle length regardless of activation level. *Eur. J. Appl. Physiol.* **116**, 1267–1277. (doi:10.1007/s00421-016-3381-3)
2. Barclay CJ. 2015 Energetics of contraction. *Compr. Physiol.* **5**, 961–995. (doi:10.1002/cphy.c140038)